# What predicts legislative success of early care and education policies?: Applications of machine learning and Natural Language Processing in a cross-state early childhood policy analysis

**Soojin Oh Park** [1]*, **Nail Hassairi** [2]

**1** Department of Learning Sciences and Human Development, College of Education, University of Washington, Seattle, WA, United States of America, **2** College of Education, University of Washington, Seattle, WA, United States of America

* parkso@uw.edu

**Data Availability Statement:** The Early Care and Education database tracks and updates weekly early care and education legislation for 50 U.S.

## Abstract

Following the pioneering efforts of a federal Head Start program, U.S. state policymakers have rapidly expanded access to Early Care and Education (ECE) programs with strong bipartisan support. Within the past decade the enrollment of 4 year-olds has roughly doubled in state-funded preschool. Despite these public investments, the content and priorities of early childhood legislation–enacted and failed–have rarely been examined. This study integrates perspectives from public policy, political science, developmental science, and machine learning in examining state ECE bills in identifying key factors associated with legislative success. Drawing from the Early Care and Education Bill Tracking Database, we employed Latent Dirichlet Allocation (LDA), a statistical topic identification model, to examine 2,396 ECE bills across the 50 U.S. states during the 2015-2018. First, a six-topic solution demonstrated the strongest fit theoretically and empirically suggesting two meta policy priorities: 'ECE finance' and 'ECE services'. 'ECE finance' comprised three dimensions: (1) Revenues, (2) Expenditures, and (3) Fiscal Governance. 'ECE services' also included three dimensions: (1) PreK, (2) Child Care, and (3) Health and Human Services (HHS). Further, we found that bills covering a higher proportion of HHS, Fiscal Governance, or Expenditures were more likely to pass into law relative to bills focusing largely on PreK, Child Care, and Revenues. Additionally, legislative effectiveness of the bill's primary sponsor was a strong predictor of legislative success, and further moderated the relation between bill content and passage. Highly effective legislators who had previously passed five or more bills had an extremely high probability of introducing a legislation that successfully passed regardless of topic. Legislation with expenditures as policy priorities benefitted the most from having an effective legislator. We conclude with a discussion of the empirical findings within the broader context of early childhood policy literature and suggest implications for future research and policy.

states and the territories. The database is publicly available at https://www.ncsl.org/research/human-services/early-care-and-education-bill-tracking.aspx LexisNexis StateNet", [online] Available: http://www.lexisnexis.com/en-us/products/state-net.page. These URLs and DOIs also have been included in the manuscript.

**Funding:** The authors received no specific funding for this work.

**Competing interests:** The authors have declared that no competing interest exist.

## Introduction

Following the pioneering efforts of a federal Head Start program, U.S. state policymakers have rapidly expanded access to early care and education programs with strong bipartisan support. Two decades of state prekindergarten policy was characterized by a substantial increase in enrollment of 4 year-olds, doubling its coverage from 14 to 28 percent of the population [1]. In fact, states now serve nearly 30 percent of children aged 4, twice as many 4 year-olds as Head Start, and more children than Head Start serves at all ages [1]. While massive amounts of state legislation on early care and education (ECE) have been introduced in past decades, what influences the successful passage of ECE bills has been left unexplored. Very little is known about how legislative content and success of ECE bills are related and whether the individual legislative productivity of the bills' sponsor matters.

### Developmental stages of early childhood policy

Despite public investments, the content, priorities, and effectiveness of these ECE legislations–enacted and failed–have rarely been examined. Early childhood policies progress through multiple stages before passing into law as shown in Fig 1. Applying the policy process framework [2], we conceptualized the progression of ECE bills in the following six stages: (1) problem identification; (2) agenda setting; (3) policy formulation; (4) policy adoption; (5) policy implementation; and (6) policy evaluation. We briefly define each stage of the policy process here. In the initial stage of problem identification, state legislators identify a problem situation and collect evidence to indicate the magnitude and importance of the issue and its determinants. A bill is typically introduced by an individual state legislator or by a relevant committee with jurisdiction over the topic related to the bill. During the policy formulation, state legislators develop pertinent and acceptable courses of action and consider possible policy options. Individual members of the legislature and the relevant committee can propose amendments in earlier readings of the bill. Power relationships play a critical role during this stage as they help determine the direction of the policy. The ultimate fate of the bill is decided during the stage of policy adoption or enactment–the primary interest and outcome of this study. A legislator introducing the bill garners support for a specific proposal in order to legitimize or authorize the policy. Both the Senate and the House of Representatives must approve the bill before it can be sent to the governor for signature. A bill can fail if it is not signed by the governor, voted down by the legislature, or not acted upon before adjournment of the legislature of the relevant committee. Another potential outcome of the bill during this stage is pending if it is neither failed nor enacted.

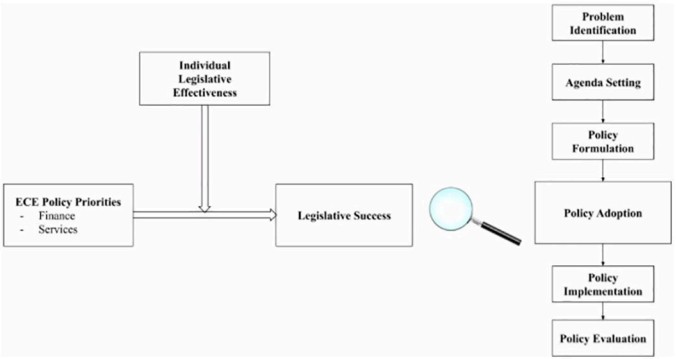

**Fig 1. Conceptual model of early childhood legislative success.**

However, most of the existing research in early childhood policy analysis has focused on the last two remaining stages–policy implementation and policy evaluation. Literature on early childhood policy implementation [3] has examined the fidelity of implementation of a curriculum intervention [4–6], teacher professional development [7–9], and coaching [10, 11] that tailor to content-focused instruction or improving classroom quality, in particular, teacher-child instructional interaction. Other scholars have examined the implementation, utilization, and impact of a child care Quality Rating and Improvement System (QRIS) on classroom quality [12] and federal or state early learning grants on underserved populations [13–15].

In the past several decades, early childhood evaluation research has established a strong evidence base on substantial and long-lasting impacts of large-scale early childhood education programs on children's learning and development [16]. Scientific evidence on the short- and long-term impacts of longstanding programs has advanced well beyond Abecedarian and the Perry Preschool programs, and Head Start [17] designed to improve lifelong chances of children in poverty [18–20]. The hallmark of the impact evaluation literature has focused on small-scale, pilot programs [21], city-wide programs [22–24], or even state prekindergarten [25–27] and federally funded programs. Rigorous efforts in cost-benefit analyses further supported public investments in high quality early learning programs [21, 22, 28], and whether such benefits were short-term [22], or long-term [21, 28].

## Policy formulation of bill content & topics as predictors of policy adoption

The policy priorities of state legislatures reflect diverse political, economic, and social interests as well as ideological preferences of policymakers and constituencies. Scholars in political science have long sought to identify factors that are associated with legislative success. The policy adoption literature suggests two models of explanatory factors that determine decisions to adopt new legislation into law. Internal characteristics models [29, 30] emphasize economic, political, and social conditions within states to explain adoption tendencies. Internal and endogenous factors such as political shifts in partisan control of state government and citizen ideology reliably predict policy priorities and legislative success [31]. On the other hand, regional diffusion models [29, 30] highlight the intergovernmental influence of one or more states on the actions of another state. Assuming geographically proximal states share more similar priorities and targeted populations, the latter model considers exogenous diffusion pressures in asserting that legislative success of a bill is determined largely by policy adoption in neighboring states. According to [29], another important element is whether the state has previously adopted other measures with similar aims or the new legislation is related to existing policies with broadly similar goals (e.g., poverty alleviation, child well-being, strengthening families). Yet these studies devote little attention to the actual content of legislation and its association with bill passage.

In this paper, we will be exploring the content of legislation through a natural language processing approach applied to the legislation content informed by an interdisciplinary review of the literature on early childhood research and public policy. Our literature review work will identify conceptually meaningful topics that could potentially inform topic assignment and validation in interpreting the results. For example, governance and finance of early childhood services are prominent subjects in legislation and function as critical mechanisms for providing equitable access to high quality services for children and their families [32]. Early childhood is a period of rapid development from conception to 8 years of age during which developmental competencies in multiple domains (i.e., cognition, language, social-emotional skills, physical and mental health) are intertwined thereby involving multiple sectors of state ECE systems. Thus, early childhood governance is one of the most complicated policy issues

[33]. ECE governance refers to the planning and development of strategies and arrangements of delivery of services. Governance mechanisms enable the allocation of responsibility of services and multiple functions within and across levels of state government. The complexities arise since early childhood governance lies at the intersection of multiple administering bodies (e.g., departments and agencies in early childhood, education, health and human services, child welfare, labor, and finance) at multiple layers of government (federal, state, local) involving various stakeholders (early learning system leaders, providers and coaches, advocates, professional development specialists). Some states jointly administer ECE programs across as many as three agencies or have two or more ECE programs each in a different agency. Joint administration by education and human services agencies is the most common form of collaboration. Many states fund and administer programs directly through a grants process that is separate from the K-12 education system. Some states have regional or local councils that share in the administration of ECE programs. Some state preschool programs are funded and administered through the public schools much like kindergarten. Governance of state prek is further complicated by the public-private delivery systems employed by most programs. Local boards and intermediate agencies may oversee policies and operations. As such, there exists an enormous variety of types of early care and education services and how they are governed. ECE programs range from center-based care, family care, Early Head Start, Head Start, state preschool and prekindergarten, and afterschool care arrangements. These programs vary widely in intended goals, levels of targeting (age groups), eligibility requirements, per child expenditure, hours per day (part-time vs. full-time), number of days offered per week, classroom size, child-staff ratios, teacher qualification, to name a few. Governance mechanisms further address questions of access, equity, and quality of provision by coordinating service delivery and supervision, optimizing allocation of resources, improving licensing procedures, provider training and workforce, regulation, and public accountability. Given the complexity, we will be expecting that the natural language processing analysis of the ECE bills uncovers governance-related priorities to ensure effective planning, implementation, and coordination. Both vertical coordination (across federal, state, and local levels) and horizontal coordination (cross-sectoral) are central to providing high quality, integrated early childhood services [32]. Recent efforts to integrate governance and financing of early childhood services in a birth-to-five framework [34] garnered broad-based, bipartisan support as these policy measures were designed to minimize fragmentation in policy implementation, improve program development and delivery, and coordinate resources and services to better support children.

In most states, prek budget is discretionary and must be fought for in each legislative session. States provide only a portion of the funding for state prekindergarten and allocate substantial funding at the local level. As a result, public resources available to fund various ECE programs require accounting of both local and state expenditures as well as fiscal governance across intermediate agencies. Differences in per child spending reflect key differences in decisions about schedules, quality, the state-local split of costs, and cost-of-living differences that make it more expensive to provide early childhood services in some states than in others [35]. Furthermore, the amount of funding allocated to ECE programs limits the number of children served, the amount of services children receive, and/or the quality of the services children receive [35]. Policymakers and program administrators inevitably face trade-offs: additional money can increase enrollments; extend program hours or days; or seek to enhance the quality through class size reductions, higher salaries, increased in-service education and coaching, or expanded services to children and families beyond the classroom. Given this extensive focus of the early childhood field on these issues, one would naturally expect the possibility of these topics occurring in the ECE legislation as well. Hence, when we later present the modeling process and employ our natural language processing algorithm, we will discuss in depth how

the extant literature informed topic assignment and validation. Little is understood regarding the associations between policy priorities such as these as reflected in legislative content and its probability of successful adoption. [36] proposed a typology for studying different types of legislation (i.e., scope and urgency) and their associations with legislative success. However, to our knowledge, very few studies have explored the association between policy content and legislative success within the ECE context.

## Moderator of policy content's effect on legislative success: Individual legislative effectiveness

Since the policy content is determined by the needs of children and families and supported by research, one would not want to sacrifice content to ensure passage. Hence, in this paper, we examine whether there are some moderators of content's effect on legislation's success, in other words whether there is another variable that could ensure passage of even difficult-to-pass policy. Rather than focusing on underlying political, economic, and organizational conditions, other political science scholars have turned attention to key characteristics of legislators. What sets apart remarkable legislators who are able to guide their bills successfully out of committee and both chambers while others are routinely met with legislative defeat? Three categories of attributes—personal, institutional, and environmental—associated with being an effective legislator may explain bill enactment [37]. First, political scientists have considered personal attributes of a legislator including education, occupation, race, gender, age, personal legislative style, personal ideology, and speech frequency that may predict legislative success [37, 38]. A second set of institutional attributes include seniority, formal position, and party affiliation. Seniority and formal positions, for example, those in the party leadership as committee chairs or ranking members, were more effective than other legislators [39]. Formal positions such as committee chairmanship imply access and control over organizational resources which leads to increased political influence. Long tenure or seniority of a legislator can rely on recognition and deference from junior colleagues, garnering their support for crafting and advocating their legislative agendas. Senior members of the legislature also have more experiences with the legislative process and lawmaking acumen into what will pass and what will fail [37]. Compared to junior, inexperienced legislators or policy opportunists who have not exhibited legislative expertise [39], senior legislators who have more electoral security are able to exercise political entrepreneurship and concentrate on legislative strategies rather than reelection strategies. Being a lawyer having years of specialized training in the theory of application of law appears to have a larger impact on effectiveness than being the committee chair [39]. Majority party status of the bill's sponsor has also been linked to bill passage [40]. A final category comprises environmental influences and constraints within which legislators must operate, including urbanicity of district or political competition. Legislators from competitive districts, for example, were less frequently considered influential and rarely gained party leadership positions. Political competition in the home district may drain considerable time and resources that would otherwise be spent in legislative policymaking. Institutional variables were found to have the strongest impact on legislative success while environmental factors were the weakest [37]. While most studies have focused on institutional and behavioral theories in explaining bill passage, [36] pointed out these models rarely consider whether the actual subject or content of a bill affects its success and illustrated why accounting for bill content in their scope and urgency may be important. Thus policy scientists in the last two decades developed Natural Language Processing (NLP) methods [41] and the usage of "text as data" [41–44] has drastically expanded in scope.

For the purpose of this study, we define "legislative effectiveness" as individual legislators' ability to successfully navigate bills to enactment independent of its underlying ideological effect. Literature on legislative effectiveness in general goes back several decades [45] and is well-reviewed elsewhere [38, 40]. Most of this work computes legislative effectiveness according to either a count or proportion of proposals that reach a certain stage of the legislative process, and tests whether some subset of independent variables accurately predicts differences in these measures across legislators. Since this pioneering effort, a number of scholars [38, 40, 46] have directly addressed the question of legislative effectiveness in the House and Senate by examining its association with institutional attributes defined above. Political sponsors may have varying degrees of influence, political leadership, and legislative specialization. These attributes include the number of political terms a legislator has served, the number of leadership roles the legislator has held, their rank in the committees they are a member of, total number of bills sponsored (productivity), and the total number of bills the legislator effectively passed (effectiveness). It seems important to make a clear distinction between legislative productivity and legislative effectiveness. Legislative productivity deals with how prolific individual legislators have been to date in considering the amount of bills they have sponsored in the past, without accounting for the number of bills that passed vs. failed. On the other hand, legislative effectiveness only takes into account only the bills that successfully passed into law. While a legislator's productivity may lead to legislative effectiveness, [38] argue that gains in legislative success become limited when legislators speak or sponsor bills too frequently.

While some argue that legislators who concentrated on a specialized area of issue activities were more successful [38], others seemed to benefit from a "shotgun approach" of introducing or sponsoring many bills on a broad array of issues. Additionally, tenure in chamber was found to be related [40, 47] and unrelated [38, 46] to differences in levels of individual legislative effectiveness. While studies offer little consensus on key ingredients of legislative effectiveness, the majority party affiliation is perhaps the most frequently considered and is found unanimously to have a positive association with legislative productivity. Studies further demonstrate that the most effective legislators were senior members of the majority party who specialized and sponsored only a few bills [40, 46]. Senior members have acquired more political experience and acumen, which may help them in "negotiating the legislative labyrinth" [40] and framing the bill in the context of broader agenda setting processes that contribute to differences in success rates. Moreover, [47] found that specialization and "legislative efficiency" increased with the number of years that members served in office. In sum, seniority and membership in the majority party play an important role in legislative effectiveness across legislative chambers at the national level as well as across time [37]. More recently, [38] have modeled legislative effectiveness as a count of the total number of bills by a member that moves through the legislative process rather than the proportion of the bills enacted.

The contribution of this paper will be in modeling the joint determination of legislative success (bill passage) by a measure of legislative effectiveness and topics identified via a combination of natural language processing topic extraction and our identification and validation of topics via expert review and content validity checks.

## Prior research on early childhood policy analysis

A common approach to analyzing policies in the field of early childhood typically relies on qualitative methods using a small sample of policies. The unit of analysis ranges from national policy documents, state standards of quality and curriculum, position statements or issue briefs. Specifically, researchers have employed various qualitative methods in conducting ECE policy analyses, including: inductive content analysis to identify themes [33, 48, 49], critical

discourse analysis of policy as text [50–52], comparative case studies of ECE policy implementation [32, 53], process evaluation of policy inputs, processes, and outcomes in addressing inequities in early learning and development [32, 54, 55]. Analysis of written texts such as legislation is an independent unobtrusive method in qualitative research [56]. While these methods are valuable for a number of reasons, qualitative analyses of a small sample of policy documents do not allow for testing associations and predictability or conducting cross-state comparisons of a large volume of text accumulated over time. We believe a machine learning approach is particularly timely in this period of statewide preschool expansion and quality improvement.

## A machine learning approach to early childhood development policy analysis

What is the main focus of thousands of pieces of ECE legislation in the past decade across the 50 states in the U.S.? Further, how does the impact of effective legislators on bill passage interact with that of different priorities of ECE bills? We seek to investigate these questions in the early childhood legislative context using a novel methodological approach of using policy text as big data. Latent Dirichlet Allocation (LDA) [57] is a generative probabilistic model for text modeling and classification which we applied to state ECE legislative data to better understand content and priorities reflected in ECE policies. LDA has been employed for big data analytics of legislative content and testing associations with other key variables in political science [58]. Methodological applications of LDA is a nascent and emerging field, and to our knowledge, no studies have examined early care and education policies in this way. Given the prominence of child care policies and early learning grants in the past decade, we present this study as a promising illustration of how massive amounts of digitized machine-readable legal text data may be analyzed to advance the future of early childhood policies and practice.

The Latent Dirichlet Allocation (LDA) method is an attempt to improve existing classification approaches [57]. The state of the art at the time was the Latent Semantic Indexing (LSI) model, bearing similarity to Factor Analysis (FA) and Principal Component Analysis (PCA). The general approach of these models is dimensionality reduction. A document-word matrix is constructed in capturing most variation (PCA) or all the common variation (FA). This "reduction of words to topics" (every word is represented in the dataset by a variable indicating how often it appears in the document) is performed using a Singular Value Decomposition. On an intuitive level, these techniques are very similar in that they "reduce dimensionality"—instead of using word frequencies as variables, we grouped the words that appear in legislative text into 'topics' or conceptual categories of policy priorities which we included for the second part of our analysis. While LSI and FA generate possibly useful results, the ad hoc way of its derivation does not help particularly form an intuitive understanding of the model or the results. In addition, FA tends to use "oblique rotation" meaning it is modeling common variation in the variables (word frequencies in our case) and PCA is designed to maximize the amount of all existing variation in the variables (words) and the components are orthogonal to each other ("orthogonal rotation"). These particular modeling choices may make these models suitable for certain applications but do not necessarily represent best modeling choice for topic modeling.

LDA model is based on a generative probabilistic model, meaning there is a complete specification of the data generating process, an attempt at modeling comprehensively how the data came about and specifying explicitly all the relevant parameters. While this difference is to some extent a matter of preference, a true test of any such method would be comparing their corresponding predictions (classifications). This is a challenging proposition to make since the

LDA, FA, and PCA are possible solutions to the problem of unclassified data. If the data was classified in the first place, we would not need them. Hence, while we can apply all of these methods and compare the topics that result from them, we would be unable to say objectively which topics are "best" [59]. With these issues in mind, we employed LDA on the grounds that it is a widely used text mining method particularly when analyzing legislation. Another indication of the validity of this approach would be whether after having identified the 'topics' we can find a statistically significant correlation between 'topics' and the probability of passage (*RQ1*). Topic Modeling [57] is a rapidly emerging area of research in educational data mining. Text analysis methods in education research have been used to 1) measure latent dispositions such as attitudes and beliefs of learners and instructors [60]; 2) explore the underlying topics and topic evolution spanning the 50-year history of educational leadership research literature [61]; 3) microclassroom processes such as MOOC interaction data [62]; 4) policy implementation and reform strategies [63]; or 5) identifying 'topics' of multiple setting-level predictors of student's language and math achievement outcomes [64]. Despite the promising potential of applying topic modeling in a variety of fields in the social sciences, its scalable, algorithmic approach to large-scale text data has received little attention among early childhood and education policy scholars [61]. To our knowledge, this study represents the first application of NLP and machine learning approaches to studying state ECE legislation (using LDA or any automated text classification method for that matter). This gives a new way of exploring and analyzing the government record and, further, gives a useful predictor of government behaviors that ultimately shape early experiences and development of children.

## Research questions

This paper employs Natural Language Processing (NLP) and machine learning approaches to examine 'text as data' drawn from ECE bills in its least understood stage in a bill's life to understand key topics of the bill that predict passage in themselves as well as in interaction with attributes of the bill's sponsors. Our study sets out to demystify the black box of "policy adoption" in the policy process framework [2] and further shed light on the hidden components that successfully facilitate bill enactment. We present a conceptual model of this current study in Fig 1 displaying hypothesized linkages among endogenous and exogenous variables as determinants of legislative success.

1. What are the most prominent policy priorities ("topics") across state ECE bills?

2. Does policy priority ("topic") predict the legislative success of the bill?

3. Does the primary sponsor's legislative effectiveness moderate the relation between policy priority and legislative success of ECE bills?

## Method

### Dataset

There are various sources of legislation data available (LegiScan, for example). Such sources usually focus on all legislation which is advantageous for political science researchers who are usually interested in the political process in general, however, we are interested in the early childhood legislation in particular. We could have opted to sample such a database by means of keyword search to narrow down the legislation to examples associated with early childhood care and education. However, this could have generated a sample containing examples that are only tangentially related to ECE or would require much more data cleaning to make sure we were including relevant legislation. The dataset originated from the Early Care and Education

(ECE) Bill Tracking. Instead, we have opted to use Early Care and Education (ECE) Bill Tracking Database maintained by the National Conference of State Legislatures (NCSL), as they have been carefully curated by the NCSL staff to include only bills relevant to the ECE issues and community. This database tracked and provided digitized machine-readable full-text of ECE legislations from the 2015-2018 legislative sessions for all 50 states in the U.S. NCSL staff curated this selection of early childhood bills from the broader universe of all state legislation. The database also provided information on [1] topical categories (e.g., 'Early Childhood Financing', 'Early Childhood Governance,' 'Prenatal, Infants and Toddlers,' 'Prekindergarten and School Readiness'); [2] states; [3] year introduced; [4] status (adopted/failed/pending), [5] primary sponsor, and [6] their party affiliation. We have downloaded all available records (9,272 in total) from the NCSL tracking database using a web-scraping program.

The data exhibited features of multi-level structure introducing possible dependence between the residual regression error term within bills introduced in the same year, by the same legislator, or in the same state. This motivated the use of the Hierarchical Linear Model (HLM) [65] for the analysis, however, since the outcome in *RQ3* was a binary variable (i.e. whether the legislation was adopted or failed), we used the Hierarchical Generalized Linear Model (HGLM) [66], which accounted for spatial and temporal correlations while using logistic function to model the outcome.

## Sample

The initial sample consisted of 9,272 bills spanning 2008 to 2018 from the most recent legislative sessions by performing a universe sampling (since the web-scraping program allowed for automation of data collection and the NCSL does not limit how many queries can be made from a single computer within any given amount of time). However, we engaged in a series of steps to address issues such as [1] removing bills that "die" due to lack of action, opposition or shifting priorities; and [2] the presence of missing key information in the data. Some of the bills, for example, did not contain the legislation text. Some other bills did not satisfy our inclusion criteria (i.e., state-level legislation) thereby excluding federal legislation, legislation from territories such as Puerto Rico or legislation from Washington, DC. The dataset also contained redundancies—bills appearing more than once during their deliberation process. They were added by NCSL staff at different stages of the bill deliberation process. We have removed all but the most recent version of those records since our dependent variable is the final outcome of the bill. Finally, *RQ2* and *RQ3* pertain to legislative outcomes that we define as either a success (enactment) or failure, so we have removed legislation whose status was still pending (while still including such legislation in analyses pertaining to *RQ1* for topic identification). The process of determining the final analytic sample is illustrated in Fig 2.

The final analytic sample included 3,203 unique state early childhood bills covering years 2015-2018 whose deliberations have been concluded and a definitive action has been taken regarding the ultimate fate of the bill, but only 2,396 bills had the full text available for further analysis. Based on our exploratory analysis, we concluded that the data with missing outcome represented new legislation introduced most recently and the bills with missing text were the older bills because NCSL was not including it at the time in their database. We had decided not to impute the missing data. The bias from not imputing the data in our case was that the final sample included more recent legislation, which we explicitly recognized but did not believe to be an issue. Hence, we used 3,203 to identify topics using LDA (*RQ1*). However, to answer our *RQ2* and *RQ3*, we employed the subsequent regression (HLM) on the sample with 2,396 observations (spanning the years 2015-2018) that contained all the information (columns/variables) required for the analysis (legislative text as well as the legislation

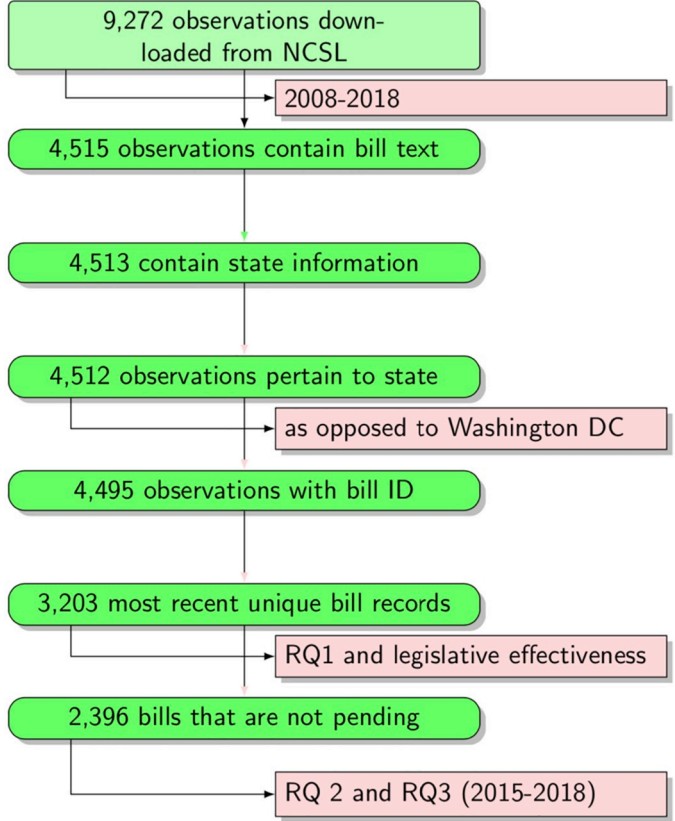

**Fig 2. Analytic samples of the study.**

outcome). The distribution of bills across the years in our sample was fairly uniform, however, some states were more active in deliberating ECE legislation than others (see Fig 3 for distribution of sample bills by state). Minnesota (373), for example, introduced the highest number of ECE bills in the most recent years followed by New York (317), Massachusetts (156), New Jersey (140), California (115), and Texas (114). Table 1 provides a description of key variables used in our study in the following order: 1) outcome, 2) explanatory variable (predictor), 3) moderator (interacts with the predictor), and 4) covariates (control). The role of covariates is to provide a more complete model and reduce bias and variance on estimates of the predictor's association with the outcome. The first column in the table gives the name of the variable used and the role in parentheses. The second column describes variable type, characteristics, and definitions. The last column provides information about the range of values. Table 2 provides additional summary statistics for these variables (mean, standard deviation, minimum, maximum). We can see in the table that 17.6% of bills in the sample were passed into law (this number is higher than an equivalent for generic legislation success reported by the political science literature). Legislative effectiveness has a very skewed distribution with most state legislators having low numbers of ECE bills they led into passage and a small number of legislators demonstrating much higher levels of effectiveness than the average legislator. The bills on average contained 42.4% of 'Child Care' topic, followed by 'PreK' (21.5%), 'HHS' (14.1%), 'Expenditures' (8.7%), 'Revenues' (7%), with 'Fiscal Governance' having the lowest proportion (6.4%) represented in the actual content of bills. At the same time, as the minimum and maximum values indicate, there were bills focusing either

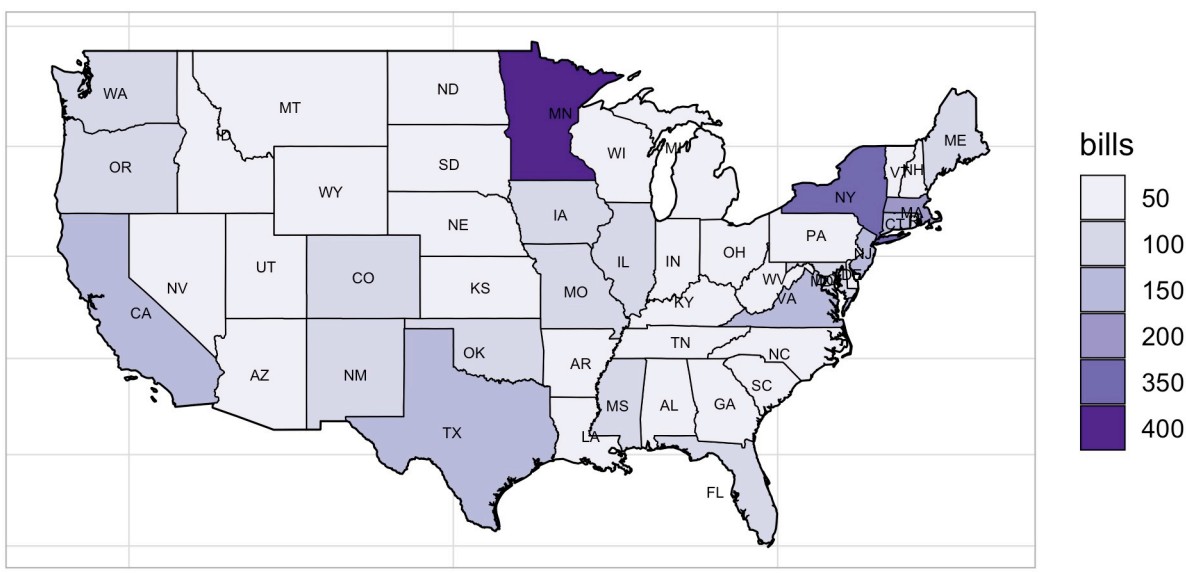

**Fig 3. ECE bills deliberated by state (2015-2018).**

exclusively on a certain topic or completely excluding them, so the dataset contained a wide range of bills.

## Text pre-processing

Prior to feeding the text of the legislation into the LDA algorithm, we have taken several steps to extract the most helpful words from the text and remove redundant ones. We excluded

**Table 1. Variables used in our analysis classified into outcomes, predictors, and controls and followed by their description and range of values they attain.**

| Variable | Description | Range |
|---|---|---|
| Passage (OUTCOME) | Binary variable indicating whether the bill passed or not (so the value 0 indicates failed bills). This variable was created by consolidating the original values "adopted", "enacted", "failed", "failed–adjourned", "pending", "pending–carryover", "to congress", "to governor", and "vetoed" into two categories–"failed" or "adopted" for addressing RQ2 and RQ3. Pending bills were removed prior to RQ2/RQ3 analysis. | Value is 1 if bill was adopted, 0 if bill failed. |
| PreK (PREDICTOR) | Continuous variable indicating how much of the PreK related topic content was present in a given legislation upon performing an LDA. | Values range between 0 to 1 (0-100%). |
| Child Care (PREDICTOR) | Continuous variable indicating how much of the Child Care related topic content was present in a given legislation upon performing an LDA. | Values range between 0 to 1 (0-100%). |
| HHS (PREDICTOR) | Continuous variable indicating how much of the HHS related topic content was present in a given legislation upon performing an LDA. | Values range between 0 to 1 (0-100%). |
| Revenues (PREDICTOR) | Continuous variable indicating how much of the content in a given legislation was related to Revenues upon performing an LDA. | Values range between 0 to 1 (0-100%). |
| Expenditures (PREDICTOR) | Continuous variable indicating how much of the content in a given legislation was related to Expenditures upon performing an LDA. | Values range between 0 to 1 (0-100%). |
| Fiscal Governance (PREDICTOR) | Continuous variable indicating how much of the content in a given legislation was related to Fiscal Governance upon performing an LDA. | Values range between 0 to 1 (0-100%). |
| LegEff (PREDICTOR; MODERATOR) | State legislators' effectiveness–the total number of bills an individual legislator shepherded as a primary author and led to passing into law at the time the bill was introduced. | Integer, ranges from 0 to 15. |
| Year (COVARIATE; random effect) | The year in which given legislation was introduced. | 2015-2018 |
| State (COVARIATE; random effect) | Categorical variable indicating the state, in which the legislation was introduced. | All 50 states were represented. |

**Table 2. Descriptive statistics of key variables used in our analysis.**

| Variable | n | Mean | SD | Min | Max |
|---|---|---|---|---|---|
| Passage | 2,396 | .176 | .381 | 0 | 1 |
| LegEff | 2,396 | .538 | 1.248 | 0 | 15 |
| PreK | 3,203 | .215 | .272 | 0 | 1 |
| ChildCare | 3,203 | .424 | .291 | 0 | .998 |
| HHS | 3,203 | .141 | .252 | 0 | 1 |
| Revenues | 3,203 | .07 | .165 | 0 | .981 |
| Expenditures | 3,203 | .087 | .165 | 0 | .921 |
| FiscalGovernance | 3,203 | .064 | .133 | 0 | .984 |

common "stop words" such as 'the' 'and' 'or' 'because' which could belong to any topic and as such do not contribute to the algorithm's primary purpose of classification of text into topics (list of stop words eliminated is available from authors upon request). In general, we have removed all prepositions, conjunctions, interjections, and articles. Additionally, we have intervened to remove some grammatical features not relevant for topic classification such as declension and conjugations—in English this comes down to removing 's' from plural form of noun and 's' from third person singular of a verb. Hence, we consolidated words that had the same root meaning (e.g., 'children' becomes 'child'; 'preschools' becomes 'preschool'). Thirdly, we removed proper names of states, cities, etc. These are a subset of proper nouns, although we have kept some as some proper nouns do have relevance in the Early Childhood Education space. For example, the Department of Health and Human Services is a proper noun with relevance to topic classification since certain matters in ECE are handled by this department while others are handled by the Department of Education (which we have also preserved). We also removed all the numbers, which we found irrelevant to topic classification. Some of these were references to other pieces of legislation, years, dates, etc.

Finally, we fixed typos when we were able to find them and removed infrequent and obviously (based on our knowledge of ECE) irrelevant words such as 'acupuncturist' or 'ammoniated' in an effort to provide a better signal-to-noise ratio to the LDA algorithm to work with (we keep a list of such words so that researchers interested in replicating our results may do so reliably). The topics identified with those extra words were not fundamentally different but rather a bit more 'noisy' or 'blurry' in a sense that there was some overlap between different topics. The LDA algorithm works best with large datasets (Nay [58] used 70,000 bills compared to our 3,203), hence we felt it may be helpful to ease its way into our text data using these pre-processing steps. While removing words such 'ammoniated' is to some extent a subjective decision, we do not claim that the topics we have discovered are 'true' or 'unique' in any sense. If one was to add 10 new pieces of legislation to the dataset, the topics discovered by LDA will be slightly altered. What we do claim is that the topics we identified through a combination of objective methods and subjective professional judgement have statistically significant relationship to the probability of passage of legislation and interact with other predictors of legislation success. In doing so, we demonstrate that standard predictors of legislation have a differential impact on the legislation's success depending on the topic of the legislation and this should be taken into account when considering strategies for promoting legislation related to the welfare of children and education of the nation.

## Analytic strategy

Our primary methodological objective was to analyze the content of the legislation and study associations between policy priorities in topically coded legislative texts and the odds of bill

adoption. Our empirical strategy extended beyond existing approaches to analyzing legislative content of policy text used in the ECE literature in several ways. First, this strategy employed a computational analysis of policy texts [41] as big data rather than qualitative coding and discourse analysis of a small number of policy documents typically used in the ECE literature. This allowed us to scale up such an analysis from dozens to thousands of pieces of legislation, across time, on a national scale. Second, using our dataset, we obtained for every piece of legislation how many 'topics' it contained and their relative proportion. Using this information in addition to data on whether the bill was enacted or not, we were able to test the associations between the bill content and its odds of adoption. Finally, we used state legislators' effectiveness as a moderator for bill content to further explore this relation. In other words, what role does early childhood legislators have in whether a bill with specific content successfully passes into law or not? Do more experienced legislators with a strong track record have a higher chance at legislative success given the topic of the bill? Consequently, we describe below our analytic strategy for addressing three resulting research questions. We employed R software version 3.4.2 [67] for all the data processing and analysis described in this paper. For *RQ1*, a library developed for R called "topicmodels" was used to perform LDA for *RQ1* [68]. For *RQ2* and *RQ3*, we used the Linear Mixed-Effects Model (lme4) package [69] to apply the HGLM model to our data.

**RQ1: What are the most prominent policy priorities ("topics") across state ECE bills?**
We used Latent Dirichlet Allocation (LDA) [57] to analyze the text of 2,396 early care and education bills. As one of the text mining methods, LDA was appropriate for algorithmically identifying recurring patterns of language from unstructured policy text data [43, 70]. Machine learning approaches to data mining find patterns and regularities in the unstructured data or extract semantically meaningful information. LDA is a generative probabilistic three-level hierarchical Bayesian model for collections of discrete data. This modeling approach allowed analysis of patterns in legislative data to better understand and conceptualize its content. In particular, it identified groupings of words across various pieces of legislation that formed the 'topics,' which we refer to as policy priorities in the context of this study. The LDA model estimated the following: 1) latent constructs or topics of policy content across all bills, 2) the probability that a certain word was associated with a certain topic, and 3) the proportion of topics in any given piece of legislation. The algorithm produced an absolute probability that a certain word was associated with a certain topic (non-exclusive) and relative odds that a word belonged to one topic rather than another. The topic weights representing the proportion of topics in legislative texts were created as an aggregation of the frequencies of those individual words and their affiliation with topics. As an example, let us say that the algorithm determined that the word 'interaction' belongs to topic 1 with probability 70% and the word 'emotional' belongs to topic 1 with probability 30%. Then imagine that there is a piece of legislation consisting of only those two words. Then the algorithm will determine that this piece of legislation is 50% composed of topic 1. This example illustrates the nature of the aggregation of information from words to documents, trading technical accuracy for simplicity of presentation. In reality, there are many words and they belong to several topics with various probabilities and the aggregation is not a simple average but rather an integral over a region of probability space of a probability density function.

However, one of the limitations of LDA is that it arrives at topic classification through a frequency analysis outlined above while human approach to topics is based on semantic structures. There is no guarantee that the frequency analysis approach the LDA takes will derive theoretically meaningful constructs without human aid [63]. To reconcile the machine and human approach, topic validation becomes essential for automated text analysis since ultimately the topics identified through LDA will be used in a narrative shared with fellow humans

(the research community that will read this paper in our case). Additionally, the LDA is not designed to infer how many topics there are. It required us to specify the number of topics. One could see this as an advantage, since it allows us to summarize the text to a level of granularity or detail chosen by us. If seen in this light, one could say that LDA 'allows' us to specify the number of topics. In our case, LDA allowed us to choose a topic classification that bears most relevance for legislation's success. One could see topic definition to be guided as the age of children served, in which case the topics would be: 'infant/toddlers', 'preschoolers', etc. Another classification could have been by activity: 'childcare' (emphasis on care), 'preschool' (emphasis on education), etc. LDA's guiding approach is frequency of words, which could be aligned with one such semantic classification but does not have to. We as researchers chose a classification provided by LDA that also had such a semantic element to it.

Existing literature in early childhood policy broadly identified policy priorities relevant to 'governance,' 'finance,' of 'programs and services' [71]. These categories of early childhood policy priorities as represented and discussed in the literature are broadly defined and do not discuss more nuanced and specific policy priorities or strategies state legislatures might propose for systemic implementation. For *RQ1*, we aimed to discover early childhood policy priorities that conformed to the issues of 'governance', 'finance', and 'programs and services' as described above. More importantly, we improved prior studies that did not as thoroughly validate the topics derived from text analysis. While there is no universal standard to guide such decision making in assigning and validating topics, automatic content analysis may generate unreliable or invalid measures [43, 72]. Thus in the current study, we aimed to use both substantive and statistical evidence to conduct comprehensive validation for topics by 1) employing a series of models that specified 2 to 7 topics, 2) using fit statistics to aid our model selection as illustrated in Fig 4, and 3) finally, evaluating the content validity and coherence of policy topics that represented conceptually and practically meaningful themes.

Fully Automated Clustering (FAC) is a classification method used to simultaneously estimate the categories and then classify documents into those categories. Thus, we applied FAC to explore emerging topics and validated the output of a 6-topic classification model based on both substantive and statistical evidence. We then carefully read and coded the textual context for each set of the 50 high frequency words that were associated with each of the six topics to further establish meaning in guiding our analysis. In labeling each topic and inferring the commonality across these words, we conducted content validity checks of six topics containing high frequency words (Fig 5) that exclusively appeared in a given topic. In addition, we carefully assessed the intent and meaning of these words in the context of legislative text, examined consistent and varying examples of how these words were used in the legislation, and finally, how these words provided a conceptual basis and coherence for each of the topics. This subjective evaluation led to determining that the 6-topic model was optimal. In the results section, we illustrate the process and results for determining topics, establishing content validity and coherence of topics, and testing their predictive validity.

**RQ2: Does policy priority ("topic") predict legislative success of ECE bills?**   In addressing *RQ2*, we employed the Hierarchical model [65]. The multi-level nature of our dataset with certain groupings of observations—legislation sponsored in the same year, in the same state, or by the same legislator which may share common characteristics not captured by regressors or covariates—prevented us from using an Ordinary Least Squares (OLS) regression, since one of the fundamental assumptions of such a regression approach is independence of all observations, which for the reasons mentioned above, would not be satisfied. If OLS was used for this type of data, the statistical significance of the regression results might be overstated, an issue which is not relevant in the LDA itself since statistical hypothesis testing is not being done for the identification of the topics themselves. Since our outcome was a binary variable (whether

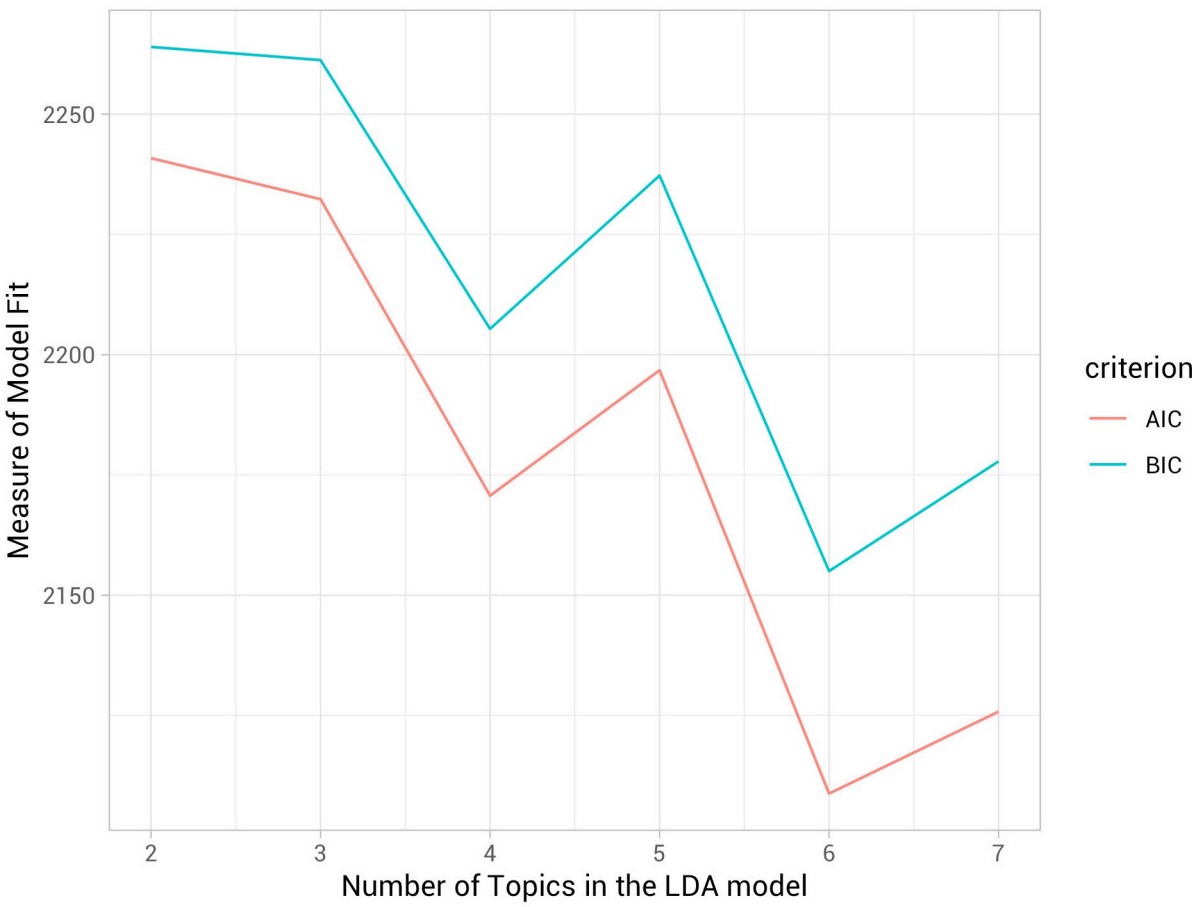

**Fig 4. Comparison of model fit of LDA with different number of topics.**

the bill passed or not), we used a Generalized Hierarchical model (logistic regression version of the hierarchical model), rather than the standard (linear) variety. We fitted the hierarchical logistic model for successful passage of the bill as a function of the six key policy priorities identified in our dataset:

$$
\begin{aligned}
Pr(\text{bill passed}) = logit^{-1}( \\
\beta_1 PreK + \beta_2 ChildCare + \beta_3 HHS+ \\
\beta_4 Revenues + \beta_5 Expenditures + \beta_6 FiscalGov \\
)
\end{aligned}
$$

The parameters of interest were $\beta_1, \beta_2, \beta_3, \beta_4, \beta_5$, and $\beta_6$, which measured the association between legislation's proportion of a given topic and its probability of bill passage. If a given beta coefficient was positive, this would mean that bills with a higher proportion of the topic associated with this parameter were more likely to be enacted and vice versa. If any given ($\beta_1$, ..., $\beta_6$) was equal to zero, then this would indicate that there was no association between odds of passage and the proportion of the relevant associated topic (this could also indicate that the LDA has not been very successful at classifying bills by topic).

**RQ3: Does the primary sponsor's legislative effectiveness moderate the relation between policy priority and legislative success of ECE bills?** To set the stage for addressing *RQ3*, we

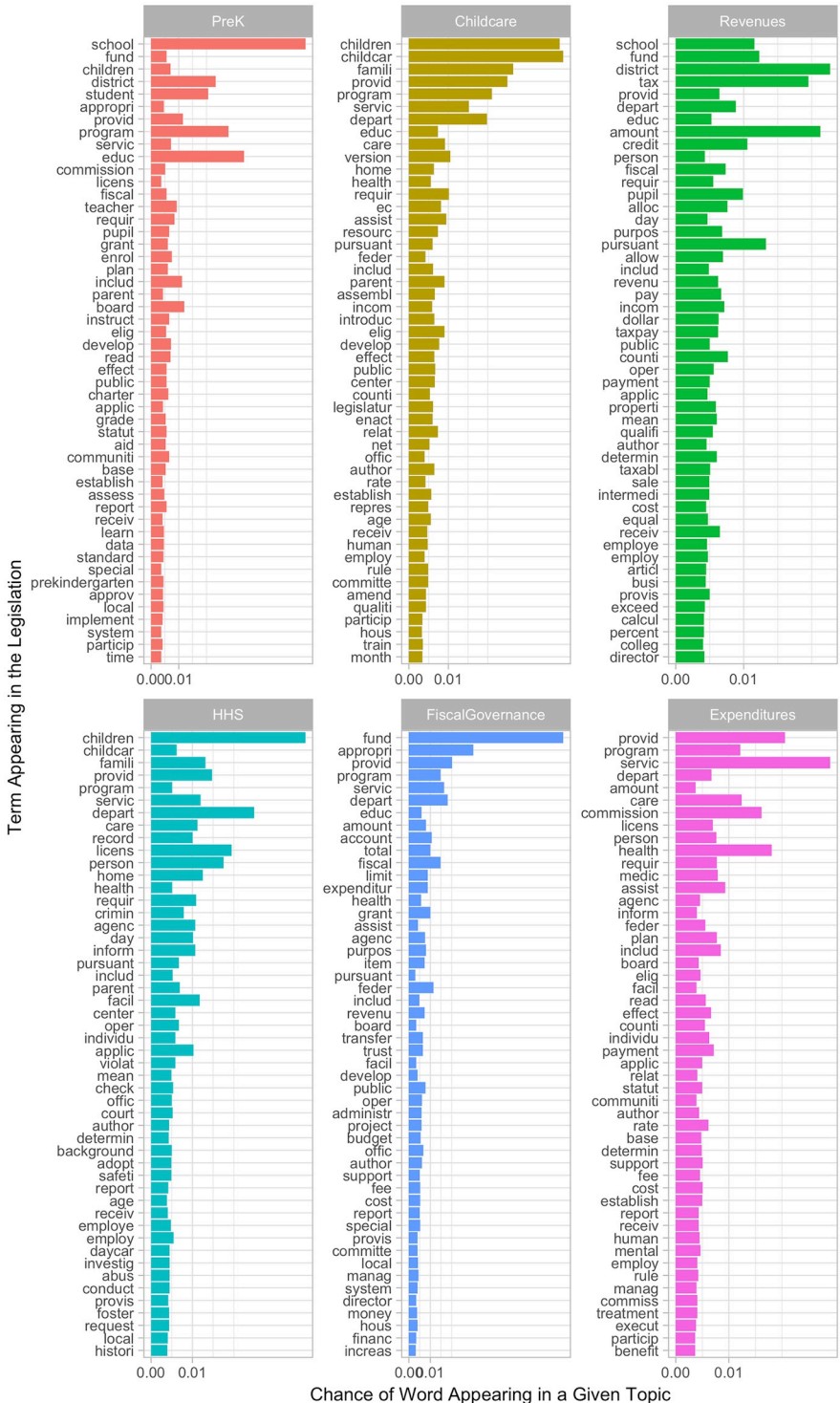

**Fig 5. Six topics discovered through LDA with 50 highest frequency words.**

first examined whether individual legislative effectiveness of state early childhood legislators alone was predictive of bill passage (*RQ3a*):

$$Pr(\text{bill passed}) = \alpha + \beta LegEffectiveness$$

Subsequently, we fitted a hierarchical logistic model for successful passage of the bill as a function of the six key policy priorities (as identified in *RQ1*) moderated by individual legislative effectiveness (see Fig 5 for visual representation of the results from this analysis). Our model was as follows:

$$Pr(\text{bill passed}) = logit^{-1}($$
$$\beta_1 PreK + \beta_2 ChildCare + \beta_3 HHS+$$
$$\beta_4 Revenues + \beta_5 Expenditures + \beta_6 FiscalGov+$$
$$\beta_7 PreK \times LegEffectiveness + \beta_8 ChildCare \times LegEffectiveness+$$
$$\beta_9 HHS \times LegEffectiveness + \beta_{10} Revenues \times LegEffectiveness+$$
$$\beta_{11} Expenditures \times LegEffectiveness + \beta_{12} FiscalGov \times LegEffectiveness$$
$$)$$

The parameters of interest were $\beta_7, \ldots, \beta_{12}$. If these parameters were equal to zero, then topics representing policy priority content in the legislation and their sponsor's legislative effectiveness may be both relevant in their own right in predicting the enactment of the bill, however, in such a case there would be no interaction effect, legislative effectiveness would not be a meaningful moderator of the relationship between policy priority and bill's odds of passage. If the parameters $\beta_7, \ldots, \beta_{12}$ were larger than zero, then a bill covering ECE policy priorities authored by a highly effective legislator (with a track record to prove it) would have a greater chance of passing successfully into law compared to a less effective legislator. If the beta parameters were negative, then being an effective legislator would make it more difficult to successfully introduce legislation containing ECE policy priorities.

## Results

Our first goal was to examine variations in ECE bills being adopted (legislative success) and individual legislative effectiveness of primary sponsors. Descriptive statistics are provided in Table 1. Of the 3,203 bills on early childhood care and education, about 18 percent in our sample successfully passed into law (mean = 0.176; sd = .38). Clearly, legislative success is difficult to achieve: While the average number of bills a member of the s tate legislature sponsored and passed into law was only 0.5, on average, the most effective legislator shepherded a total of 15 ECE bills through adoption in years 2015-2018. Of the 2,396 bills containing full text, the average proportion of policy content focusing on PreK was 21.5%, Child Care 42.4%, HHS 14.1%, Revenues 7%, Fiscal Governance 6.4%, and Expenditures 8.7%. In general, content related to 'programs and services' represented a higher proportion relative to content related to 'finance.' Furthermore, political sponsors varied widely in their individual legislative effectiveness: A total number of ECE bills a state legislator was able to pass successfully ranged between 0 and 15 bills over the lifetime of our dataset (2015-2018).

### RQ1: Key policy priorities in state early childhood bills

We present *RQ1* findings that resulted from first specifying the number of topics through a series of LDA models and using fit statistics to aid our model selection, and then evaluating the content validity and semantic coherence of policy topics. Drawing from policy text as big

data, we aimed to identify the most prominent key 'topics' or policy priorities that constitute early childhood legislation that were considered during the most recent legislative sessions (2015-2018) for 50 states. We employed a series of models that specified 2 to 7 topics. First, a two-topic solution initially suggested two broadly defined topics we had identified in the literature review: 'finance' and 'services.' However, topic allocations were more precisely estimated and resulted in more specific policy priorities when using the six-topic solution. As displayed in Fig 4 and Table 3, a six-topic solution demonstrated the strongest fit theoretically and empirically based on BIC and AIC measures of model fit. However, the "best" model needs to capture the topics of interest to the researcher [73, 74]. Model choice is typically based at least partially on subjective considerations similar to those in the qualitative research [43]. These six topics represented more nuanced and conceptually meaningful topics that belonged to the two cross-cutting 'meta-topics': 'ECE finance' and 'ECE services.'

Having determined a final set of six policy topics, we further elaborate on the process of assigning the meaning of the topics and assessing their content validity. Topic validation becomes essential for automated text analysis since the LDA algorithm's underlying frequency analysis may identify 'word groupings' as 'topics', however, these 'word groupings' may not strike a human observer as having a semantic structure. To ensure computer-generated topics captured topics that were theoretically and practically meaningful, two expert coders evaluated the content validity and semantic coherence of the six topics. Content validity denotes the extent to which the topics identify coherent and distinct sets of ECE policy priorities and measure conceptually sound constructs. Having expert coders who have relevant theoretical and practical knowledge was critical for assessing topics' content validity [63]. Fig 5 displays a unique set of 50 words with highest probabilities of appearing in each topic and that best distinguish the topics from one another. Using these high probability terms, we independently coded and labeled each of the six topics and assessed its coherence (how these words hold together as a construct) and the extent to which a given topic is meaningful and consistent with the literature. To further aid this process, we carefully read high probability terms in original legislative text, inferring its definition, meaning, and usage and their relation to the topic. Upon repeating this process for each model specification, we compared and discussed our qualitative coding decisions. We reached agreement on classifying most of the topics, and refined the label for just one topic, 'Fiscal Governance,' that integrated both the administration and financing of ECE services.

For ECE finance, we identified three specific topics—'revenues', 'expenditures', and 'fiscal governance'. Unique keywords with highest probabilities of appearing under 'revenues' were adjustment (of schedule to facilitate department accounting or unit allocation), agreement (i.e., property tax or tobacco tax agreement), a llocation, allowable tax credit, allowance (for instructional classroom support) business, calculation, corporation, credit, distribute, dollar, income, fiscal intermediary, properties, residence, sale, tax, taxable, taxpayers. These words together suggested budgetary sources needed to finance early childhood care and educational services, particularly in schools and districts as seen in Fig 5. High frequency words for 'expenditures' included benefit, care, children, clients, disabled children, families, hospital, maltreatment, medical, medicaid, mental health, nurse and treatment, suggesting what it would cost for various providers in delivering early childhood services including mental health and social welfare services for direct beneficiaries. Furthermore, several keywords, for example, indicated expenditures in terms of financing the early childhood workforce (i.e., structuring salaries and benefits). This process involves setting standards for teacher qualifications and other structural characteristics (e.g., class size, ratio, hours per week) of programs that affect cost. Since public expenditures on state preschool come from federal, state as well as local sources, states vary in their program schedule (half-day or a full-day) and leave the schedule up to local discretion

**Table 3. Models predicting bill passage using LDA for topic classification with different numbers of topics.**

|  | 2-topics | 3-topics | 4-topics | 5-topics | 6-topics | 7-topics |
|---|---|---|---|---|---|---|
| ECEFinance | −0.90** |  |  |  |  |  |
|  | (0.30) |  |  |  |  |  |
| ECEServices | −0.63* |  |  |  |  |  |
|  | (0.28) |  |  |  |  |  |
| topic1 |  | −1.02*** | −1.17*** | −1.24*** |  | 1.13 |
|  |  | (0.30) | (0.31) | (0.30) |  | (0.77) |
| topic2 |  | −0.43 | 0.18 | −0.90* |  | −2.48*** |
|  |  | (0.29) | (0.32) | (0.39) |  | (0.44) |
| topic3 |  | −0.86* | −1.44*** | −0.99** |  | −0.80 |
|  |  | (0.38) | (0.36) | (0.35) |  | (0.42) |
| topic4 |  |  | 0.61 | 0.08 |  | 0.18 |
|  |  |  | (0.45) | (0.31) |  | (0.35) |
| topic5 |  |  |  | 0.55 |  | 0.69 |
|  |  |  |  | (0.46) |  | (0.53) |
| PreK |  |  |  |  | −0.36 |  |
|  |  |  |  |  | (0.36) |  |
| ChildCare |  |  |  |  | −1.72*** |  |
|  |  |  |  |  | (0.33) |  |
| Revenues |  |  |  |  | −2.73*** |  |
|  |  |  |  |  | (0.52) |  |
| HHS |  |  |  |  | 0.56 |  |
|  |  |  |  |  | (0.36) |  |
| Expenditures |  |  |  |  | 1.60*** |  |
|  |  |  |  |  | (0.48) |  |
| FiscalGovernance |  |  |  |  | 0.90 |  |
|  |  |  |  |  | (0.50) |  |
| topic6 |  |  |  |  |  | 1.90*** |
|  |  |  |  |  |  | (0.53) |
| topic7 |  |  |  |  |  | −1.35*** |
|  |  |  |  |  |  | (0.33) |
| AIC | 2240.86 | 2232.33 | 2170.68 | 2196.77 | 2108.76 | 2125.80 |
| BIC | 2263.99 | 2261.24 | 2205.37 | 2237.24 | 2155.02 | 2177.84 |
| Log Likelihood | -1116.43 | -1111.17 | -1079.34 | -1091.38 | -1046.38 | -1053.90 |
| Num. obs. | 2396 | 2396 | 2396 | 2396 | 2396 | 2396 |
| Num. groups: state | 52 | 52 | 52 | 52 | 52 | 52 |
| Num. groups: year | 4 | 4 | 4 | 4 | 4 | 4 |
| Var: state (Intercept) | 1.74 | 1.76 | 1.87 | 1.80 | 2.18 | 2.05 |
| Var: year (Intercept) | 0.15 | 0.15 | 0.17 | 0.16 | 0.18 | 0.20 |

*** $p < 0.001$,

** $p < 0.01$,

* $p < 0.05$

[35]. Local authorities determine how thinly state funding is stretched regarding length of day and frequently, data are unavailable about how many children are served with each type of schedule. In managing state budget—both revenues and expenditures—the third finance-related topic emerged as 'fiscal governance' dealing with account, appropriation, balance,

capital (capital gain, capital expenditure, capital loss), commission, committee, contract, finance, item, local, money, officials, reimbursement, salaries, schedule, fiscal transfer (by administrative bodies to accounts), and trust (governing board of trustees as well as trust management). These words seemed to suggest the logistics of locating the money to pay for the programs at the legislative level and facilitating efficient distribution and use of resources. A commonly shared set of words that appeared across the three topics further confirmed ECE finance as the underlying theme—amount, authorize, communities, cost, counties, department, fund, payment, provide, and public.

For ECE services, three specific topics included governance and provision of early childhood services: 'prekindergarten (PreK),' 'child care,' and 'health and human services (HHS).' We found that three key topics—'prekindergarten' (preK), 'child care', and 'health and human services' (HHS)—addressed policy questions of what (types and quality of provision, program components, and dosage), for whom (targeted population, access and participation trends particularly among historically marginalized children and their families), where (delivery settings), as well as by whom (administration leaders and providers). Classification of these three topics closely reflect the complexity of early childhood governance as well as the fragmentation and duplication of federal funding streams and state administration across early childhood programs and services. More specifically, unique words with highest probabilities of appearing for 'preK' included academic, achieve(ment), charter, college-ready, commission(er), district, enroll, grade, instruct, kindergarten, prekindergarten, preschool, pupil, school, student, and teacher. In comparison, high frequency words under 'child care' included assist(ing), children (rather than pupils or students under 'preK' topic), counties, daycare, income, infant, subsidi (-es, -ze), and home visit. These two topics, 'preK' and 'child care', mutually focused on promoting the development and early learning opportunities for eligible children through educational programming and services. Unlike the topics 'preK' and 'child care', 'HHS' embodied a broader system of early childhood and social welfare services as suggested by words such as a buse, adopt(ion), agency, child care, convict(-ed, -ion), court, criminal, daycare, foster care, health, home visit, neglect, personnel, for example. The theory of change undergirding 'HHS' services posits that preparing children for success in school and life involve not only their cognitive development and academic enrichment but also their physical and mental health as well as social-emotional well-being and family support services. Types of services may include home visits, parent support or training, referral to social services, and health services. Additionally, examining a commonly shared set of words underlying these three topics further supported our classification of these topics having to do with ECE programs and services: Words such as authorize, children, parents, program, provide, public, purpose, receive, require and services further supported our conceptualization of these three topics as complementary yet distinct dimensions of providing public early childhood programs and services that are designed to promote children's learning and healthy development.

## RQ2: Legislative success in relation to policy priorities

We then included the quantitative measures of policy topics derived from *RQ1* as predictors of the outcome of our interest: the odds of bill passage. While our content validity and semantic coherence analysis above was aimed to ensure internal validity of the topics revealed by the LDA, in this section *RQ2* provided an opportunity to examine whether these topics actually bore relevance to the legislative outcomes. Hence, this step also served as an external validity check on the relevance of the topics uncovered by LDA. Overall, as we can see in Table 4 (model *RQ2*) and Fig 6, legislation topic was significantly correlated with a probability of passage for the following topics: 'ChildCare' ($\beta_2 = -1.72$, $p < .05$), 'Revenues' ($\beta_4 = -2.73$,

**Table 4. Final model results for RQ2 and RQ3.**

|  | RQ2 | RQ3a | RQ3b |
|---|---|---|---|
| PreK | −0.36 |  | −1.66*** |
|  | (0.36) |  | (0.44) |
| ChildCare | −1.72*** |  | −2.32*** |
|  | (0.33) |  | (0.39) |
| Revenues | −2.73*** |  | −3.55*** |
|  | (0.52) |  | (0.71) |
| HHS | 0.56 |  | −1.08* |
|  | (0.36) |  | (0.46) |
| Expenditures | 1.60*** |  | −2.43** |
|  | (0.48) |  | (0.87) |
| FiscalGovernance | 0.90 |  | −1.64* |
|  | (0.50) |  | (0.80) |
| (Intercept) |  | −1.84*** |  |
|  |  | (0.31) |  |
| LegEffectiveness |  | 1.47*** |  |
|  |  | (0.08) |  |
| PreK:LegEffectiveness |  |  | 1.34*** |
|  |  |  | (0.24) |
| ChildCare:LegEffectiveness |  |  | 0.67*** |
|  |  |  | (0.14) |
| Revenues:LegEffectiveness |  |  | 1.71** |
|  |  |  | (0.55) |
| HHS:LegEffectiveness |  |  | 2.30*** |
|  |  |  | (0.33) |
| FiscalGovernance:LegEffectiveness |  |  | 2.76*** |
|  |  |  | (0.71) |
| Expenditures:LegEffectiveness |  |  | 6.69*** |
|  |  |  | (0.88) |
| AIC | 2108.76 | 1672.10 | 1501.12 |
| BIC | 2155.02 | 1695.22 | 1582.07 |
| Log Likelihood | -1046.38 | -832.05 | -736.56 |
| Num. obs. | 2396 | 2396 | 2396 |
| Num. groups: state | 52 | 52 | 52 |
| Num. groups: year | 4 | 4 | 4 |
| Var: state (Intercept) | 2.18 | 1.10 | 1.53 |
| Var: year (Intercept) | 0.18 | 0.24 | 0.28 |

*** $p < 0.001$,

** $p < 0.01$,

* $p < 0.05$

$p < .05$), and 'Expenditures' ($\beta_5 = 1.6$, $p < .05$). Hence, on average, the higher the proportion of 'ChildCare' and 'Revenues' topics in the legislation, the lower the probability of passage (other things held constant), while the 'Expenditures' topic proportion in the legislation was positively associated with the probability of passage of the average piece of legislation. For illustration, we calculated the odds of passage for legislation consisting of only one topic for each of the six topics: .7 for 'PreK', .18 for 'ChildCare', .07 for 'Revenues', 1.8 for 'HHS', 5 for

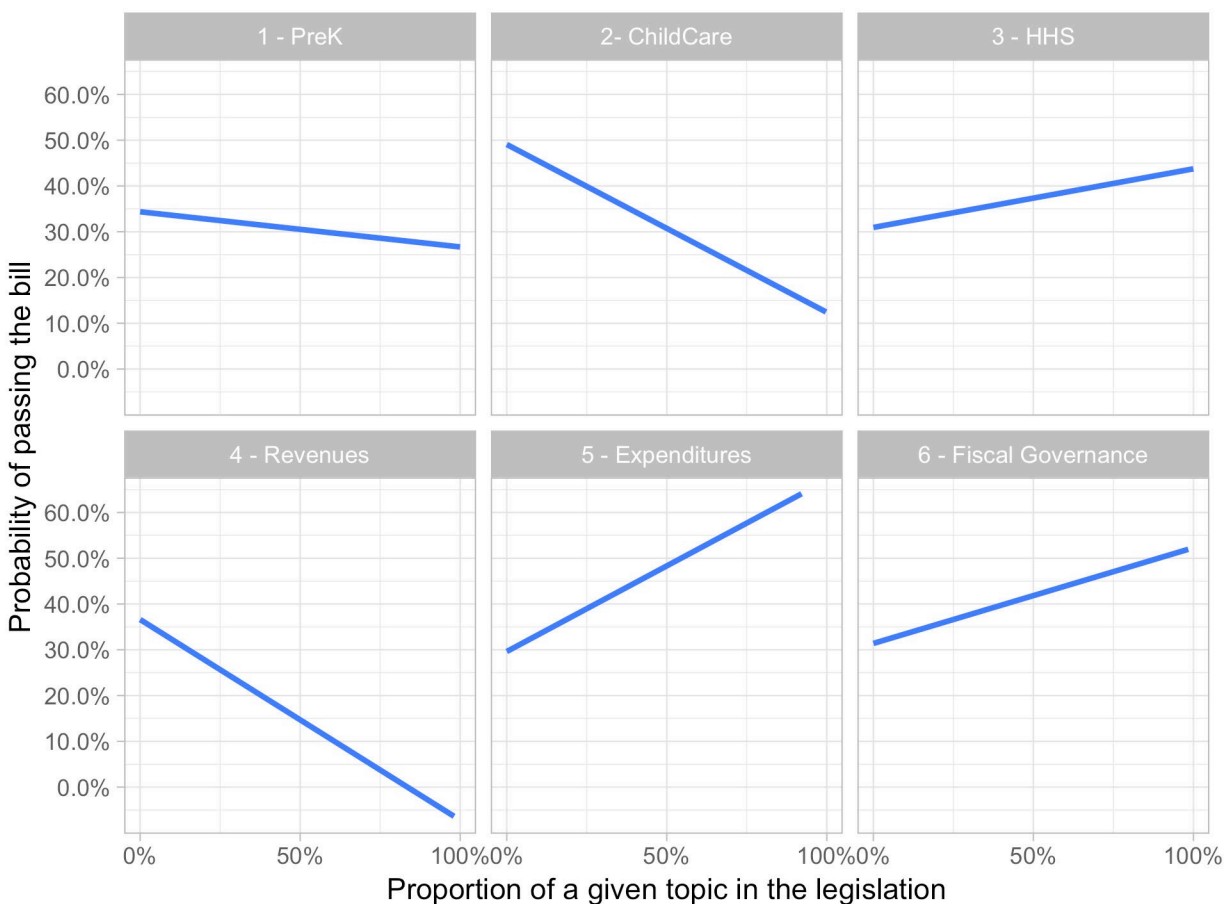

**Fig 6. Bivariate association between 'topics' (bill content) and the odds of the bill being adopted.**

'Expenditures', 2.5 for 'Fiscal Governance'. This implies that the odds are in favor of policy priorities related to 'HHS', 'Expenditures', and 'Fiscal Governance' while being against policy priorities supporting 'PreK', 'ChildCare', and 'Revenues.' For example, the odds of 5 to 1 for 'Expenditures' is defined as:

$$\frac{Prob(passage)}{Prob(failure)} = 5$$

In other words, the legislation is 5 times more likely to succeed than fail if it is focused exclusively on 'Expenditures'.

## RQ3: Relation of legislative effectiveness to policy priorities and legislative success

The third goal of our study was to understand which additional factors (other than the policy priorities) were most important in predicting a bill's success and whether there is any interplay between legislative priorities in the legislative text and characteristics of sponsoring policymakers. Specifically, we examined whether the primary sponsor's individual legislative effectiveness predicted the legislative outcome and moderated the relation between policy priorities and legislative success (see Fig 7 for an illustration of the hypothesized mechanism).

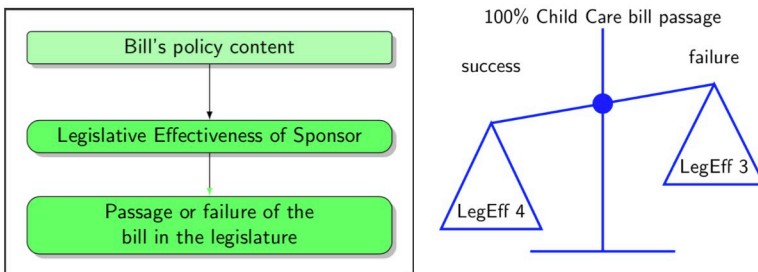

**Fig 7. Legislative effectiveness moderated the relationship between the bill's content and its legislative outcome.**
Scales indicate that for a hypothetical bill with its entire content focused on ChildCare, for example, a highly effective sponsor who successfully passed 4 bills would tip the balance in favor of passage (right).

First, we found that lawmakers with a higher level of individual legislative effectiveness were more likely to shepherd the bill to a successful adoption compared to their peers whose records proved them to be less effective (see Fig 8 and Table 4). Table 4 presents the regression results for model *RQ3a*: The intercept was −1.84 which represented the log odds of passage of the legislation. To get the actual baseline odds we exponentiated this to obtain .16 (approximately 1 in 5 odds of passage). Since the coefficient on legislative productivity was $\beta = 1.47$, the

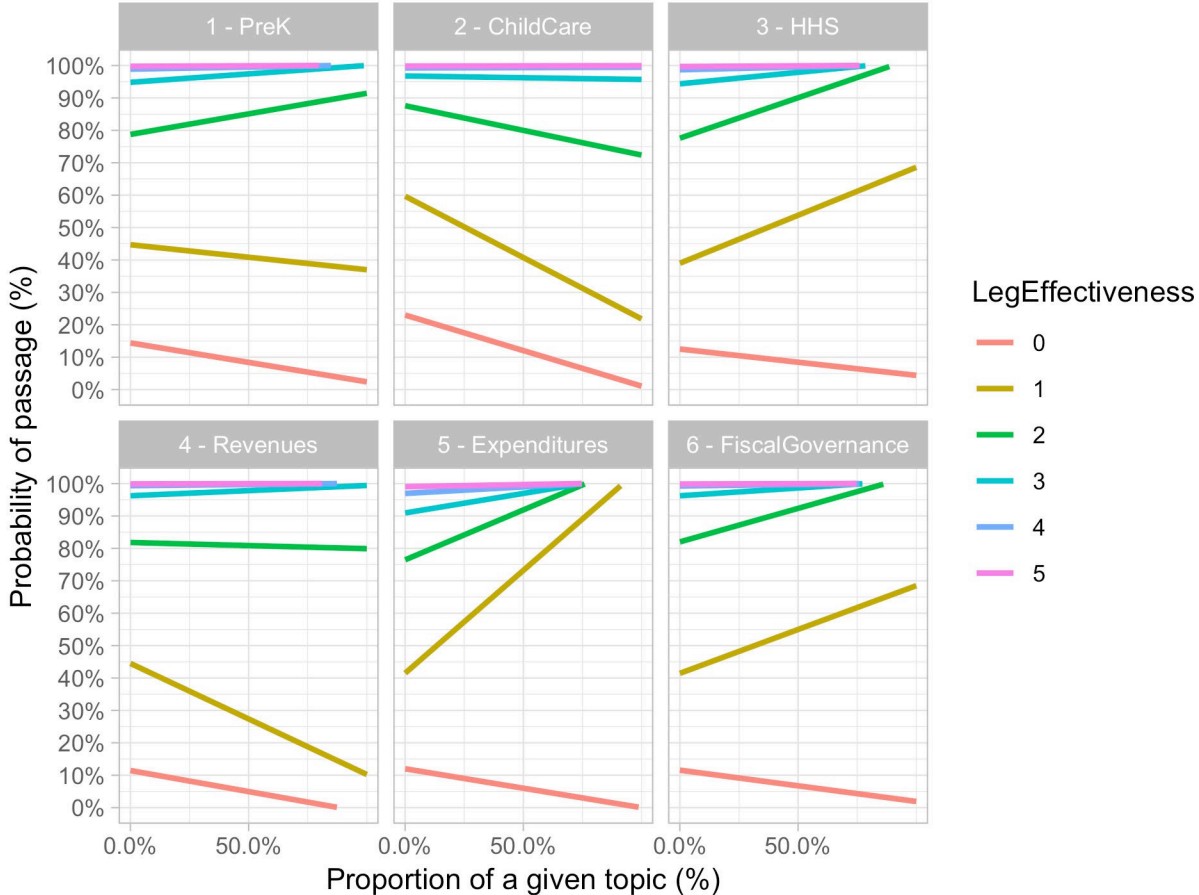

**Fig 8. Moderating effect of legislative effectiveness on the association between the proportional coverage of a policy priority and the probability of bill passage.**

log odds of a legislator with one successfully passed bill was

$$\log\left(\frac{Prob(passage)}{Prob(failure)}\right) = \alpha + \beta \times 1 = -1.84 + 1.47 = -.37$$

If we exponentiated this again, we obtained .7 (approximately 3 in 4 odds of passage). We can see that between a legislator with no record of success and a legislator with a record of one successfully passed piece of legislation, the odds they would pass the next ECE bill nearly quintupled, a substantial difference in a policy context. Still, the odds for such a legislator were slightly against them since the odds were smaller than 1, meaning the probability of failure was still higher than the probability of passage (the odds are a ratio of the probability of success and the probability of failure). Starting with a legislator with two successfully passed pieces of legislation, the log odds

$$\log\left(\frac{Prob(passage)}{Prob(failure)}\right) = \alpha + \beta \times 2 = -1.84 + 2 \times 1.47 = 1.1$$

began to be in favor of passage. After exponentiation, we obtained an estimate of 3 for the odds themselves. Hence, such a legislator had odds 3 to 1 in favor of passage, three times more likely to succeed than fail.

However, as observed in Fig 7 and Table 4, the interaction between legislator effectiveness and bill content was somewhat complex. Highly effective legislators (5 or more bills passed before) had an extremely high probability of successfully introducing legislation regardless of policy topic (possibly because they also do not introduce legislation with low chances of success). Legislators with zero track record of successful legislative activity, on the other hand, had a very low probability of successfully introducing a bill of any policy topic (in the 10-20% range depending on topic). Legislation with 'Expenditure' policy priorities benefitted the most from having an effective legislator as a sponsor. Another way of looking at this interaction was to ask how many successfully passed bills a sponsor needed to have to turn a bill with 100% of a certain policy topic from odds against passage (lower than 1) to odds in favor of passage (higher than 1)—from a situation where it was less likely to pass than fail to a situation where it was more likely to pass than fail (akin to balancing the scales in Fig 7). For certain policy priorities only one successfully sponsored legislation under the sponsor's record gave them such favorable odds: 'HHS', 'Expenditures', and 'Fiscal Governance'. If we call this number of previously passed bills required to have odds in favor of passage a 'breakeven point', then for the 'HHS', 'Expenditures', and 'Fiscal Governance' this breakeven point was one successfully passed bill. Hence, we could say that for 'HHS', 'Expenditures', and 'Fiscal Governance', the legislative effectiveness breakeven point was 1, since a single legislative achievement of the sponsor takes us from a situation where the scales were tilted against success to a situation where they were skewed in favor. However, for the remaining topics, it required a higher level of legislative effectiveness from the potential bill's sponsor. The following were the 'breakeven points' for the remaining topics: 2 ('PreK'), 3 ('Revenues'), and 4 ('ChildCare'). Hence, certain ECE policy priorities required at least an intermediate effective legislator ('PreK', 'Revenues', and 'ChildCare'), while other policy priorities provided potential opportunities for early, less experienced legislators with just one successfully sponsored bill to improve their legislative record ('HHS', 'Expenditures', and 'Fiscal Governance').

## Sensitivity analysis

We conducted extensive sensitivity analyses to confirm the robustness of our findings. Three limitations of our study include (1) issues of causality, (2) sensitivity of results to the possible variations in the output of the LDA procedure, and (3) sensitivity of results to the choice of the legislative effectiveness variable instead of legislative productivity.

First, we use an observational sample rather than experimental or quasi-experimental data. We have attempted to obtain all the available data, however, in this particular case, this meant all the legislation curated by the NCSL staff. NCSL may have a selection criteria that were not publicly known that may have affected the external validity of our results. We encourage researchers to obtain ECE legislation from other sources to corroborate our results. Consequently, since our sample was observational rather than experimental in nature, we were not making any causal conclusions. Rather, we made claims about observed associations in the data and speculated as to their possible meaning. For example, in regard to the association between legislative effectiveness and passage, there could be two scenarios that would generate the same correlation, however, with opposite causality. The first scenario that makes more sense to us could be where bill content determines bill passage. The second scenario could suggest that people have certain expectations or opinions regarding which topics are more likely to fail and they do not even introduce such bills. Hence the topics of the bills introduced are determined by their chance of passage. So the causality runs the other way. At the same time, the nature of the activity made it rather unlikely to test the hypotheses in question experimentally: Legislators do not introduce legislation with random content just to see whether it passes or fails.

One may wonder whether the addition of more legislative text data would have changed the topics identified and to what extent. LDA performs better on larger datasets and we have aimed to collect as large a dataset as possible, however, there was always a chance that with more legislation, the algorithm would produce different topics than we have identified in our study. We have modified some criteria of the topic generating algorithm during text pre-processing to see whether this would radically alter results, and this was not the case. The literature on LDA suggests removing general, irrelevant, or redundant words such as 'this', 'that', 'because', or 'the', for example, that would conceivably appear with high frequency in any topic. Some of these removals fell in a gray area and we used our own discretion informed by existing literature, practical knowledge of the state early childhood system, and meaningful interpretations of data. In such cases, we have run the algorithm with and without such words to determine whether their presence or absence substantially altered the results. We have also tried removing suffices and keeping only word stems so that words such as 'belong' and 'belonging' were categorized as the same word. We have run the LDA algorithm for both stemmed and un-stemmed words. Different approaches inevitably produced very slightly different lists of high frequency words but topic classification remained consistently stable.

Additionally, our moderator variable—legislative effectiveness—could have possibly served as a proxy for some other legislator characteristics such as political capital, seniority, majority party membership, reputation, charisma, ruthlessness, ability to close deals, or ideological flexibility. To probe the robustness of our variable of choice, we have created and analyzed various legislative experience variables of the primary sponsor (4 variables) across individual legislative effectiveness—(1) total bills authored over the time frame of our data; (2) total bills authored by the time the bill in question was authored, (3) total successful (passed) bills over the time frame of our data, and (4) total successful bills by the time the bill in question was authored. All of these variables were correlated with each other and statistically significant predictors of bill passage. While productivity (i.e., a sheer quantity of bills sponsored) may primarily allude

to the passion and political will, effectiveness may encompass additional factors that were critical for shepherding the bill towards enactment. In the end, aligned with extant literature, individual legislative effectiveness showed the strongest association with the odds of passage in our analyses, corroborating our choice.

## Discussion

Can big data predict which early childhood state legislation will be more likely to pass into law? What are the most prominent policy priorities in early childhood state legislation, and what are the odds of these priorities for bill adoption? The extant literature has barely scratched the surface on these important questions. Integrating perspectives across child care and early education policy, political science, natural language processing, machine learning, and computational linguistics, this was the first large-scale study of early childhood state legislation using the Latent Dirichlet allocation (LDA) for topic modeling. Natural Language Processing (NLP) methods such as LDA employ classification algorithms to process very large unstructured datasets in automated ways, informed by theoretical concepts, to make real-time predictions. We synthesize key findings of the study and suggest several possible paths for this research to advance along.

One of the key contributions of this study was our conceptualization of the key topic predictors which were empirically supported by big data of ECE legislation. Results demonstrated that early childhood legislation prominently focused on two broad areas of policy priorities: early childhood finance and services. Under each policy area, we identified three key 'topics' theoretically informed and empirically supported by state legislative data. For early childhood finance, three priorities centered around 'revenues,' 'expenditures,' and 'fiscal governance,' meaning policies were put in place to set the course for where the funding will come from, how the fiscal resources will be allocated to local levels, and how the fiscal mechanisms will be governed. When designing state preschool programs, one of the most difficult decisions facing policymakers involve setting requirements related to program structure: These cost-related decisions include first deciding appropriate age and dosage (i.e., hours per week), then staff qualifications and compensation, class size and staff-to-child ratio [35]. For early childhood services, three prominent 'topics' were: 'prekindergarten,' 'child care,' and 'health and human services,' representing complex governance mechanisms for coordinating across the comprehensive and diverse gamut of available programs publicly funded to support young children, particularly those from historically underserved communities in concentrated areas of poverty in the U.S. These three service-focused topical priorities further reflect the complex history of fragmented, inefficient, and unnecessarily complex federal and state early childhood policies and programs created in response to crises and changing goals, to produce a complex terrain for early learning that leaves out many of the nation's vulnerable children in Black, Indigenous, Latinx, and other communities of color [71, 75]. State policymakers have attempted to integrate incoherent federal funding streams with growing city- and state-funded early childhood programs. Policymakers have gotten locked into choosing among three bad options: tinkering around the edges of existing programs, cutting them, or adding new components on top of what exists [76]. It remains to be seen how state early childhood legislation in the coming years will reflect shifting trends and needs for integrated, multisectoral services in the context of birth-to-five system alignment, coordinated governance, blending-braiding funding streams, integrated data systems, compensation parity and retention. State and federal ECE policies have yet to tackle the most pressing equity priorities to dismantle systemic racism and other forms of oppression in ECE such as providing equitable access to high quality professional training and supports for early childhood service providers, ensuring equitable dissemination

of public funds and resources, embedding equity in monitoring and accountability systems, eliminating harsh discipline that disproportionately affect Black children, expanding access to dual language immersion approaches, addressing racial equity in early intervention and special education access, identification, and inclusion, for example [75].

Additionally, we found that a bill's text alone has predictive power. We constructed a measure of bill content as a proportional coverage of policy priorities found in legislative text and tested whether or not the proportional coverage of these topics was related to legislative success, which we defined as the odds of bill passage.

It is interesting to compare ECE legislation to legislation more broadly. While federal bills failed 96 percent of the time [58], close to 18 percent of ECE bills in our sample were passed into law, which was more than 4 times the success rate of passing for an average federal bill. It is possible that state legislatures represent more socially cohesive constituencies than the Congress of the United States (bills in this study are state legislation whereas Nay worked with federal data [58]). The most efficient Congress of the past two decades, the 106th, pushed 6 percent of bills to enactment. In an era of such political divisiveness at the federal level, bipartisan support for early childhood state legislation has provided a "critical period" of policy window when "hot bills" or bills that catch the political tide may be able to ride a wave of political interest and broad public support into legislative success [77]. In the case of Washington state, Ruth Kagi, a former senior member of the state legislature, had the highest level of legislative effectiveness among all state legislators during 2015-2018. Her legislative effectiveness in championing landmark legislation further reflects the strong public-private partnerships, local philanthropic and state advocacy efforts, and most importantly, broad-based support and public will to advance racial equity and educational justice in early learning in Washington. ECE bills, on average, with a higher proportion of content covering 'Health and Human Services', 'Expenditures', and 'Fiscal Governance' were more likely to pass relative to bills focusing largely on 'PreK', 'Child Care', and 'Revenues'. Applying a topic modeling approach proposed by [78], future studies can predict voting patterns based on the contents of bills using roll call data—historical records of legislators' votes on a set of issues. Furthermore, central to the complex sociopolitical process of lawmaking is language. Framing of language, for example, used in drafting the bill by using certain words like 'impact' and 'effects' increased the chances for climate-related bills in the House whereas 'global' or 'warming' spelled trouble [58]. An emerging area in early childhood policy has examined a significant role of framing language in increasing the chance of successful passage by using qualitative methods such as strategic frame analysis [79, 80], content analysis of news media, cognitive interviews and focus groups to discern deeply held beliefs and assumptions of the general public, peer discourse analysis of inter-group negotiations around social issues [81], and explanatory metaphor development [82, 83], and field frame analysis [84]. Other text data that could benefit from NLP methods include transcripts of legislative debates, press releases, policy proposals, early learning performance standards, or topic extraction using social media data given that tweets related to early childhood education have rapidly grown on Twitter. Lastly, the field of early childhood policy may benefit from a text reuse approach [85] proposed "text reuse methods" to systematically trace the progress of shared policy ideas that might advance across a series of individual bills and how they move through the legislative process.

Prior studies have examined how individual-level behavior and institutional variables associated with legislators affect bill passage. We thus explored whether a key characteristic of state legislators (i.e., a strong legislative track record) may be an important factor in addition to policy content. We found that state legislators' effectiveness was a strong predictor of bill passage, and had a moderating influence on the early childhood bills' chance of passage in particular to the extent that if a bill has long odds (uphill battle) of passage, recruiting a successful legislator

with a record to boot, could improve those odds considerably. This would imply that the ECE community should not give up some of its pressing policy priorities just because the perceived baseline chance of passage may be low, but rather seek to recruit a champion capable of altering the odds of passage in favor of the policy priorities. Highly effective legislators who previously passed five or more ECE bills had an extremely high probability of sponsoring their legislation to enactment regardless of topic. Further, for each topic we have computed a 'break-even point', a minimum number of bills that a legislator must have passed (legislative effectiveness), to be an effective sponsor for a legislation with such a topic (make the odds of passage more favorable than not; see Figs 6 and 7). Results indicated that while bills containing a greater proportion of 'PreK', 'Child Care', and 'Revenues' required at least an intermediate effective legislator since these topics were more difficult to pass, other bills focusing on 'Health and Human Services', 'Expenditures', and 'Fiscal Governance' provided potential opportunities for early, less experienced legislators with just one successfully sponsored bill to improve their legislative record. These findings on differential effects of legislative effectiveness on the association between bill content and passage further aligned with Weissert's [39] conceptualizations of policy entrepreneurs who serve as legislative experts in a specialized policy area and policy opportunists who have not demonstrated expertise and persistence but have sponsored the bills associated with an issue at the time when the policy window opens, subsequently benefitting from enhanced visibility and standing [39]. Moreover, the committee system encourages legislative specialization, rewarding members who become policy experts in a particular area and provide reliable, trustworthy information to fellow members in state legislatures. An earlier study of 70,000 bills introduced in the U.S. Congress from 2001 to 2015 identified two most important factors in predicting a bill's success: sponsors in the majority party and sponsors who served many terms [58]. Interestingly, we did not find any significant association between the party affiliation of the bills' sponsor and the probability of passage. Other key characteristics of state legislators beyond individual legislative effectiveness that enhance or derail ECE bill passage will generate interesting directions for future research.

Quantitative political science literature further attests that the most effective legislators were senior members of the majority party with a legislative specialization [40, 46]. Building on this literature, our study makes several important contributions to the literature on legislative success and early childhood policy analysis. We point out several existing theories that may drive the interaction effects detected in the study. We concur with Hibbing [47] that members of Congress are not "mere automatons" whose legislative hopes are dictated solely by institutional factors beyond their control. Rather, as politicians pursue different goals, consider institutional constraints, and take advantage of political opportunities [86], they make assessments about the long-term chances of their proposals and set about choosing activities that might improve those chances. Our results demonstrated that selectively effective legislators were able to overcome institutional impediments and political barriers in seeing their legislative agendas to fruition. Legislators must navigate time constraints, shifting political climate and priorities, constituency pressures, and the difficulty of mapping legislative solutions onto specific issues for targeted populations of young children and families. The costs and benefits of bill sponsorship particularly in terms of staff time and resources or the reputational costs associated with consistently introducing losing bills, for example, must be considered when analyzing a legislator's pursuit of public policy agendas [38, 87].

Furthermore, the literature is not conclusive regarding the relation between increased bill sponsorship (legislative productivity) and legislative effectiveness. Moore and Thomas [46] found that increasing sponsorship activity significantly decreased legislative effectiveness for senators, especially those of the majority party. Overly prolific legislators may find a decreasing return from the additional time and staff effort dedicated to legislating. Frantzich [40]

suggested that a broad legislative approach results in a "double payoff" whereas less prolific and more focused legislators succeeded less frequently. Our finding of the importance of legislative effectiveness on bill passage challenges this notion and further supports previous literature on legislative specialization. Those legislators who were more likely to narrow their legislative scope have a higher chance of passing their sponsored bills. Our dataset did not allow us to test the associations of seniority, political leadership, and majority party membership with the individual legislative effectiveness of state legislators. However, we observed that Ruth Kagi, mentioned earlier, with the highest level of legislative effectiveness was in fact a senior member of the state legislature and belonging to a majority party in the state of Washington.

## Promise and limitations of big data in predicting the future of early childhood policy

Politics of legislative policymaking occur in the context of written and spoken word [43]. While scholars have long recognized the importance of language in understanding and predicting the complex process of lawmaking, the massive costs of analyzing collections of policy texts have hindered their use in political science research. LDA is a widely used method within topic modeling that allows for discovering latent patterns or abstract 'topics' that emerge across unstructured big data that would otherwise be prohibitively costly in terms of RA time or prone to bias and a variety of other cognitive heuristics that interfere with objective research. Our analysis of ECE legislation via automatic text classification algorithm generated salient topic groupings that predicted bill enactment, which allowed making inferences that were previously impossible. The evaluation of the topic models using text mining methods is still an active area of research and lacks widely accepted evaluation methods. Formal comparison of predictive effectiveness between LDA and Factor Analysis has not been done to date [59]. Unlike factor analyses that model shared variance between variables, LDA identifies components with the highest between-class variance.

LDA may be a promising topic modeling tool in analyzing and predicting which state ECE legislative bills will likely pass thereby better informing the public discourse, state legislative policymaking, and advocacy efforts. However, such applications of language analysis does not illuminate how the minds of state legislature works nor the process, strategy or politics of ECE. While the use of big data broadly in research on public policies and publicly-funded programs has lagged behind other fields such as computer science and medicine [88], LDA in particular has been utilized in understanding legislative policymaking [57, 58]. Additionally, the application of computer-assisted analyses of large-scale text such as text reuse methods [85], topic modeling [57] and classification methods are far sparse in education policy research [63]. However, a burgeoning body of work has applied machine learning algorithms in educational research to better understand how teaching and learning can be enhanced for whom under what conditions [60, 89]. Recent studies, for example, have built artificial neural network models that predict students' language and math performance at large scale [64], introduced various methods in educational data science (EDS) for examining students' massive open online courses (MOOCs) interactions [62], or using LDA to analyze text data from thousands of school improvement reports to identify reform mechanisms that reduced student chronic absenteeism and improved achievement [63]. As such text mining can enhance administrative data that many early childhood researchers have been utilizing for decades as well as address barriers and limitations surrounding administrative data by adding richness that is often buried in digital text without the cost of primary data collection. Compared to standard quantitative methods used in early childhood policy analysis, text mining allows for large-scale

learning with far-reaching implications and will generate ongoing policy discussions related to transparency and accessibility to open data. Computational methods have been increasingly used to analyze social media data and government information systems as well. Natural Language Processing methods hold promise to improve the operations of early childhood agencies as well as provide meaningful insights for rulemaking and policymaking towards transforming the workforce and continuously improving services for children and families. While automated content analysis methods can substantially reduce the costs and time of analyzing massive collections of political texts, these methods cannot replace the careful reading and thoughtful analysis by the researcher [43] but rather complement user-centered, context-sensitive approaches to qualitative research [63]. In the process of building integrated data systems or large datasets, we argue that policy goals, administrative priorities, and well-grounded understanding of state early learning systems are critical for guiding such developments rather than being led by technological advances alone. Our approach integrates both the machine learning approach for the discovery of topic classification as well as the traditional approach to qualitative coding and validation of those topics. Recognizing the limitations of text modeling methods, this paper helped us envision how automated content analysis methods guided by careful thought and reasoning have a promising potential to revolutionize the field of early childhood policy.

## Directions for future research

State legislative policymaking in early care and education has remained vigorous for the past decades due to significant expansions of the early learning landscape that varies tremendously from state to state. Cross-state similarities and differences in the provision and expenditure on early childhood development programs continue to grow. States vary in their governance structures, approaches to inter-agency collaborations and public-private partnership, program standards, ancillary services, monitoring and accountability practices, and fiscal challenges and availability of existing infrastructure and resources [90]. Given all of this variation in policies, it is hardly surprising that there is a great deal of variation in how and how much state early childhood programs are funded. All of this variation raises concerns that access to high quality early learning will be highly unequal across and within states. However, the tremendous variation among states also provides fertile ground for research on policy and on the consequences of different policies for program administration, classroom practices, and children's learning and development [35].

Future studies can build on policy diffusion literature to investigate any evidence of policy borrowing among states that share geographical proximity. Policy diffusion literature suggests that diffusion within a federal political system can be varied with states learning from one another (horizontal diffusion), the national government may emulate successful state policies (bottom-up, vertical diffusion), or states may choose to enact policies that go beyond national standards [30]. Diffusion studies offer many insights to guide the examination of underexamined policies in early care and education. Evidence of policy borrowing among states is an important area of concern as states may replicate ineffectiveness and discourage innovative breakthroughs much needed in the high-stakes political context of early childhood programs. While it can be very constructive and effective in some situations, under pressing time and political pressure for accountability, state policymakers and early learning leaders engage in horizon scanning and look at other contexts and peer states to gain an informed, evaluative perspective on their own policy-making process. The rapidly changing nature of policymaking in the early learning frontier at times necessitates states to direct resources to uncritically transplant "best practices" from other contexts without first fully understanding the nature of the

problem or actual needs and priorities of the constituents. At times, ECE bill introduction could indicate compensatory legitimation—an attempt by the state to retrieve control and legitimize political authority lost in the perception of its citizen—which has been detrimental for democratic lawmaking. In such a political climate, states may fail to meaningfully invest in early childhood policies and programs by replicating rhetorical imperatives that do little to reduce systemic barriers and institutionalized disadvantages particularly for historically marginalized children and their families.

Another interesting path in advancing this research is probing whether policy borrowing is evident in legislative content among clusters of states that are systematically (i.e., mixed-delivery vs. center of school-based only, universal vs. targeted, share the same contractor of service delivery), historically (new vs. longstanding state-driven programs), or politically aligned. Macro-level contexts of early childhood policy including h istory, political climate, and scale of early care and education systems and the presence of universal prek (access and quality) or the emergence of birth-to-five comprehensive approach such as Preschool Development Grant Birth through Five, for example [34, 91] may further shape legislative content and bill enactment.

## Conclusion

Our study proposes a new analytic approach to unlocking the potential of legislative data to inform future policymaking in the early care and education frontier. Very few studies in the field of early childhood consider how policymaking occurs at state and federal levels and under what conditions state legislators achieve success in committees, on the floor, and at the enactment stage of the legislative process. We formulated computationally the prediction of which ECE bills will pass using information from the content of bills. We conceptualize policy text as an untapped potential for strengthening evidence-based policymaking and propose several extensions of the findings to build a new area of inquiry in early childhood policy. This paper illustrated how big data and machine learning approaches have the potential to examine policy-relevant questions that have previously been left unexplored. Organizations advocating for ECE legislative agendas have to choose who to approach to support desired legislation. Our work provides empirical support for the importance of choosing a primary bill sponsor with a solid track record of legislative effectiveness. Legislative process is about the art of the possible. Legislators have to decide how much of a given priority to include in legislation. Findings confirm that finance-related ECE legislation, especially regarding 'revenues' (who will pay) is a hard sell. Advocates may take such matters into account when strategizing to push for certain legislation and how much fiscal asks to include in any given bill. Findings may help guide targeted advocacy efforts by assigning the more challenging policy priorities to more senior legislators (or not intensely involving senior legislators with legislation that may be relatively easy to pass), identifying which policy priorities to push for in times or large/small majorities in the legislative bodies, or may be useful for early childhood researchers and organizations engaging in state legislative action.

## Acknowledgments

The authors are grateful to Tom Halverson for providing insightful feedback on earlier drafts of this paper. We also want to thank anonymous reviewers and editors for providing helpful comments. Additionally, we recognize the tremendous efforts made by the National Conference of State Legislatures (in collaboration with LexisNexis) for making full text of state bills on early care and education publicly available, and in particular, Alison May for providing background information we needed in building the dataset. The content and opinions are

solely the responsibility of the authors and do not represent the official views of the National Conference of State Legislatures or the University of Washington.

## Author Contributions

**Conceptualization:** Soojin Oh Park.

**Data curation:** Nail Hassairi.

**Formal analysis:** Soojin Oh Park, Nail Hassairi.

**Investigation:** Soojin Oh Park, Nail Hassairi.

**Methodology:** Nail Hassairi.

**Project administration:** Soojin Oh Park, Nail Hassairi.

**Resources:** Soojin Oh Park, Nail Hassairi.

**Software:** Nail Hassairi.

**Supervision:** Soojin Oh Park.

**Validation:** Soojin Oh Park, Nail Hassairi.

**Visualization:** Soojin Oh Park, Nail Hassairi.

**Writing – original draft:** Soojin Oh Park, Nail Hassairi.

**Writing – review & editing:** Soojin Oh Park, Nail Hassairi.

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
