## [Decision Letter · Decision Letter 0]

16 Nov 2020

PONE-D-20-13754

What predicts legislative success of Early Care and Education (ECE) policies?: Applications of machine learning and natural language processing in a cross-state early childhood policy analysis

PLOS ONE

Dear Dr. Park,

Thank you for submitting your manuscript to PLOS ONE. After careful consideration, we feel that it has merit but does not fully meet PLOS ONE’s publication criteria as it currently stands. Therefore, we invite you to submit a revised version of the manuscript that addresses the points raised during the review process.

We look forward to receiving your revised manuscript.

Kind regards,

Mingming Zhou, Ph.D.

Academic Editor

PLOS ONE

Journal Requirements:

3. We note that Figure 3 in your submission contains map images which may be copyrighted. All PLOS content is published under the Creative Commons Attribution License (CC BY 4.0), which means that the manuscript, images, and Supporting Information files will be freely available online, and any third party is permitted to access, download, copy, distribute, and use these materials in any way, even commercially, with proper attribution. For these reasons, we cannot publish previously copyrighted maps or satellite images created using proprietary data, such as Google software (Google Maps, Street View, and Earth). For more information, see our copyright guidelines: http://journals.plos.org/plosone/s/licenses-and-copyright.

You may seek permission from the original copyright holder of Figure 3 to publish the content specifically under the CC BY 4.0 license. 

If you are unable to obtain permission from the original copyright holder to publish these figures under the CC BY 4.0 license or if the copyright holder’s requirements are incompatible with the CC BY 4.0 license, please either i) remove the figure or ii) supply a replacement figure that complies with the CC BY 4.0 license. Please check copyright information on all replacement figures and update the figure caption with source information. If applicable, please specify in the figure caption text when a figure is similar but not identical to the original image and is therefore for illustrative purposes only.

Reviewers' comments:

Reviewer's Responses to Questions

**Comments to the Author**

1. Is the manuscript technically sound, and do the data support the conclusions?

Reviewer #1: Yes

Reviewer #2: Yes

2. Has the statistical analysis been performed appropriately and rigorously? 

Reviewer #1: Yes

Reviewer #2: Yes

3. Have the authors made all data underlying the findings in their manuscript fully available?

Reviewer #1: Yes

Reviewer #2: Yes

4. Is the manuscript presented in an intelligible fashion and written in standard English?

Reviewer #1: Yes

Reviewer #2: Yes

5. Review Comments to the Author

Reviewer #1: # I belive the illustration on the document-word matrix can be expanded further. Readers may wonder how authors created the matrix.

# Please illustrate further on the preprocessing of your text data.

# Please elaborate further on how authors assign the meaning of the topics. It's not clear from the manuscript.

# Authors mentioned multilevel structure of the data and used HGLM. I am just wondering whether LDA is OK for the multilevel data structure. Further, in general, is it ok to apply machine learning approach such as LDA to multilevel data structure? Why?

Reviewer #2: This paper presents an original contribution to the field through an analytic approach implementing machine-learning to unlock the potential of legislative data to inform future policymaking in early childhood-care and education.

The rationale for the research question is clearly stated. The general approach and statistical analyses were described and carried out rigorously. The discussion and conclusion appropriately derive from the data presented.

Although "the use of big data in research on public policies has lagged behind other fields", it is important to highlight some recent studies which applied machine-learning to predict educational outcomes at large scale and/or educational data mining field (for example: Musso MF, Cascallar EC, Bostani N and Crawford M (2020) Identifying Reliable Predictors of Educational Outcomes Through Machine-Learning Predictive Modeling. Frontiers in Education. 5:104. doi: 10.3389/feduc.2020.00104, and Romero, C & Ventura, S. (2017) Educational Data Science in Massive Open Online Courses. Wiley Interdisciplinary Reviews: Data Mining and Knowledge Discovery · January 2017).

6. PLOS authors have the option to publish the peer review history of their article (what does this mean?). If published, this will include your full peer review and any attached files.

Reviewer #1: **Yes: **Jae Young Chung

Reviewer #2: No

---

## [Author Response · Author response to Decision Letter 0]

27 Dec 2020

Reviewer #1a : Please illustrate further on the preprocessing of your text data.

RESPONSE : Thank you for your helpful feedback. We have added a new section “text

preprocessing” between the descriptions of the data source and our analytic strategy as follows:

Text Preprocessing

Prior to feeding the text of the legislation into the LDA algorithm, we have taken

several steps to extract the most helpful words from the text and remove redundant ones.

Prior to using the text as input for the LDA algorithm 1 , we excluded common “stop

words” such as ‘the’ ‘and’ ‘or’ ‘because’ which could belong to any topic and as such do

not contribute to the algorithm’s primary purpose of classification of text into topics. In

general, we have removed all prepositions, conjunctions, interjections, and articles.

Additionally, we have intervened to remove some grammatical features not relevant for

topic classification such as declension and conjugations--in English this comes down to

removing ‘s’ from plural form of noun and ‘s’ from third person singular of a verb.

Hence, we removed such “stem words” that have the same root meaning (e.g., ‘children’

becomes ‘child’; ‘preschools’ becomes ‘preschool’). Thirdly, we removed proper names

of states, cities, etc. These are a subset of proper nouns, although we have kept some as

some proper nouns do have relevance in the Early Childhood Education space. For

example, the Department of Health and Human Services is a proper noun with relevance

to topic classification since certain matters in ECE are handled by this department while

others are handled by the Department of Education (which we have also preserved). We

also removed all the numbers, which we found irrelevant to topic classification. Some of

these were references to other pieces of legislation, years, dates, etc.

Finally, we fixed typos when we were able to find them and removed infrequent

and obviously (based on our knowledge of ECE) irrelevant words such as ‘acupuncturist’

or ‘ammoniated’ in an effort to provide a better signal-to-noise ratio to the LDA

algorithm to work with 2 . While removing words such ‘ammoniated’ is to some extent a subjective decision, we do not claim that the topics we have discovered are ‘true’ or

‘unique’ in any sense. If one was to add 10 new pieces of legislation to the dataset, the

topics discovered by LDA will be slightly altered. What we do claim is that the topics we

identified through a combination of objective methods and subjective professional

judgement have statistically significant relationship to the probability of passage of

legislation and interact with other predictors of legislation success. In doing so, we

demonstrate that standard predictors of legislation have a differential impact on the

legislation’s success depending on the topic of the legislation and this should be taken

into account when considering strategies for promoting legislation related to the welfare

of children and education of the nation.

1 List of stop words eliminated is available from authors upon request. 

2 We keep a list of such words so that researchers interested in replicating our results may do so reliably.

The topics identified with those extra words were not fundamentally different but rather a bit more ‘noisy’

Reviewer #1b : I believe the illustration on the document-word matrix can be expanded further.

Readers may wonder how authors created the matrix. Please elaborate further on how authors

assign the meaning of the topics. It's not clear from the manuscript.

RESPONSE :

Document-word matrix and how it was created

Thank you for your question. We had a bit of a hard time understanding this particular question.

The phrase ‘document-word matrix’ doesn’t appear anywhere in the manuscript and we have

not really commented on it at all. Literal reading of your question would lead us to describe the

document-word matrix, where the document is a piece of legislation and word is a word

appearing in that legislation. This matrix would then have as many rows as there are pieces of

legislation (3,203) and a number of columns equal to the number of words in all pieces of

legislation combined. This would be a very sparse matrix as a minority of the total collection of

words appears in any particular piece of legislation. This would be a frequency matrix where the

numbers inside the matrix would represent how often a particular word appears in a particular

document. We do not operate on this matrix ourselves but rather feed this matrix to the LDA

algorithm, which operates on it. This matrix was created using the R software, whereby the

original dataset also had 3,203 rows but the columns were more of the standard variety such as

year legislation was deliberated, party of the sponsor, etc, variables described in Table 1 & 2 of

the manuscript. One of those variables is legislation text -- the whole legislation text is located in

a cell of this dataset. This column with legislative text is then broken apart into multiple columns,

each containing a single word. In practice, because this is a very sparse matrix, the data is

actually converted into a long form, where one column is the legislation ID and the other column

is a word. This long dataset has duplicates, since words can appear multiple times in the same

piece of legislation. This dataset is then tallied so that each word-document pair appears only

once and an additional column representing frequencies is added. There is not much we have

to say about this matrix (the wide version of it, which has a bit more intuitive if inefficient [in

terms of computer memory use] structure) other than that the number of columns is partly related to the number of words we filtered from the text, which is the subject of the previous

question on text preprocessing, where this step is described. The reason why we do not have

much to say about it is the same reason why we are using LDA in the first place. It is clumsy,

too large, and at the end of the day, we do not care about how words affect the passage of

legislation. Word is too small of a unit of text to carry much in a way of semantic features. This is

why the LDA essentially compresses this matrix into a document-topic matrix with fewer

columns and more meaning. In this response, we will try to describe various elements of our

process that might bear some relevance on your question as we try to guess the intent behind

your question.

Assignment of meaning to topics

We addressed this comment by incorporating substantial changes in the following sections:

literature review, RQ#1 analytic strategy, and RQ#1 results. Assigning the meaning of the topics

was driven by both theoretical and practical knowledge of the early childhood field as well as

empirical evidence. Prior to analysis, we articulated the existing body of research and

foundational theories in early childhood policy literature that informed our decisions in assigning

and validating the topics. Please refer to the literature review section--“ Predictors of Legislative

Success of ECE Bills: Policy Content & Topics .” Specifically, our systematic review explored

governance and finance of early childhood services as prominent topics in both U.S. and global

early childhood policies. We also included explicit language to communicate how the initial

literature review helped inform our labeling of the initial meta-topics: ‘finance’ and ‘services’:

In this paper, we will be exploring the content of legislation through a combination of

literature review and natural language processing. Our literature review work will

identify conceptually meaningful topics that could potentially inform topic assignment

and validation in interpreting the results produced by the natural language processing

algorithm used in this paper. For example, governance and finance of early childhood

services are prominent subjects in legislation and function as critical mechanisms for

providing equitable access to high quality services for children and their families (Britto

et al., 2014)...

Given this extensive focus of the early childhood field on these issues, one would

naturally expect the possibility of these topics occurring in the ECE legislation as well.

Hence, when we later present the modeling process and employ our natural language

processing algorithm, we will discuss in depth how the extant literature informed topic

assignment and validation.

Rather than imposing these broadly known topics a priori, we were interested in exploring

whether the prominent constructs or policy topics in ECE bills would uncover more nuanced and

understudied yet meaningful topics when using LDA. LDA is particularly useful when learning

the patterns of massive text data to identify emerging topics that are theoretically meaningful but

potentially underexamined. We have made significant revisions and expanded the RQ1 analytic

strategy and RQ1 results sections (see below) to elaborate further on topic assignment and

validation as can be seen below.

RQ1 Analytic Strategy :

We used Latent Dirichlet Allocation (LDA) (Blei et al., 2003) to analyze the text

of 2,396 early care and education bills. As one of the text mining methods, LDA was

appropriate for algorithmically identifying recurring patterns of language from

unstructured policy text data (Grimmer & Stewart, 2013; Hindman, 2015). Machine

learning approaches to data mining find patterns and regularities in the unstructured data

or extract semantically meaningful information. LDA is a generative probabilistic

three-level hierarchical Bayesian model for collections of discrete data. This modeling

approach allowed analysis of patterns in legislative data to better understand and

conceptualize its content. In particular, it identified groupings of words across various

pieces of legislation that formed the ‘topics,’ which we refer to as policy priorities in the

context of this study. The LDA model estimated the following: 1) latent constructs or

topics of policy content across all bills, 2) the probability that a certain word was

associated with a certain topic, and 3) the proportion of topics in any given piece of

legislation . The algorithm produced an absolute probability that a certain word was

associated with a certain topic (non-exclusive) and relative odds that a word belonged to

one topic rather than another. The topic weights representing the proportion of topics in

legislative texts were created as an aggregation of the frequencies of those individual

words and their affiliation with topics. As an example, let us say that the algorithm

determined that the word ‘interaction’ belongs to topic 1 with probability 70% and the

word ‘emotional’ belongs to topic 1 with probability 30%. Then imagine that there is a

piece of legislation consisting of only those two words. Then the algorithm will

determine that this piece of legislation is 50% composed of topic 1. This example

illustrates the nature of the aggregation of information from words to documents, trading

technical accuracy for simplicity of presentation. In reality, there are many words and

they belong to several topics with various probabilities and the aggregation is not a

simple average but rather an integral over a region of probability space of a probability

density function.

However, one of the limitations of LDA is that it arrives at topic classification

through a frequency analysis outlined above while human approach to topics is based on

semantic structures. There is no guarantee that the frequency analysis approach the LDA

takes will derive theoretically meaningful constructs without human aid (Sun et al.,

2019). To reconcile the machine and human approach, topic validation becomes essential

for automated text analysis since ultimately the topics identified through LDA will be

used in a narrative shared with fellow humans (the research community that will read this

paper in our case). Additionally, the LDA is not designed to infer how many topics there

are. It required 3 us to specify the number of topics. Existing literature in early childhood

policy broadly identified policy priorities relevant to ‘governance,’ ‘finance,’ of

‘programs and services’ (Kagan & Reid, 2008). These categories of early childhood

policy priorities as represented and discussed in the literature are broadly defined and do

not discuss more nuanced and specific policy priorities or strategies state legislatures

might propose for systemic implementation. For RQ1, we aimed to discover early

childhood policy priorities that conformed to the issues of ‘governance’, ‘finance’, and

‘programs and services’ as described above. More importantly, we improved prior studies

that did not as thoroughly validate the topics derived from text analysis. While there is no

universal standard to guide such decision making in assigning and validating topics,

automatic content analysis may generate unreliable or invalid measures (Grimmer & Stewart, 2013; Wilkerson & Casas, 2017). Thus in the current study, we aimed to use

both substantive and statistical evidence to conduct comprehensive validation for topics

by 1) employing a series of models that specified 2 to 7 topics, 2) using fit statistics to aid

our model selection as illustrated in Figure 4, and 3) finally, evaluating the content

validity and coherence of policy topics that represented conceptually and practically

meaningful themes.

Fully Automated Clustering (FAC) is a classification method used to

simultaneously estimate the categories and then classify documents into those categories.

Thus, we applied FAC to explore emerging topics and validated the output of a 6-topic

classification model based on both substantive and statistical evidence. We then

carefully read and coded the textual context for each set of the 75 high frequency words

that were associated with each of the six topics to further establish meaning in guiding

our analysis. In labeling each topic and inferring the commonality across these words, we

conducted content validity checks of six topics containing high frequency words (Figure

5) that exclusively appeared in a given topic. In addition, we carefully assessed the intent

and meaning of these words in the context of legislative text, examined consistent and

varying examples of how these words were used in the legislation, and finally, how these

words provided a conceptual basis and coherence for each of the topics. This subjective

evaluation led to determining that the 6-topic model was optimal. In the results section,

we illustrate the process and results for determining topics, establishing content validity

and coherence of topics, and testing their predictive validity. 

RQ1 Results:

We present RQ1 findings that resulted from first specifying the number of topics

through a series of LDA models and using fit statistics to aid our model selection, and

then evaluating the content validity and semantic coherence of policy topics. Drawing

from policy text as big data, we aimed to identify the most prominent key ‘topics’ or

policy priorities that constitute early childhood legislation that were considered during

the most recent legislative sessions (2015-2018) for 50 states. We employed a series of

models that specified 2 to 7 topics. First, a two-topic solution initially suggested two

broadly defined topics we had identified in the literature review: ‘finance’ and ‘services.’

However, topic allocations were more precisely estimated and resulted in more specific

policy priorities when using the six-topic solution. As displayed in Figure 4 and Table 3,

a six-topic solution demonstrated the strongest fit theoretically and empirically based on

BIC and AIC measures of model fit. However, the “best” model needs to capture the

topics of interest to the researcher (Roberts et al., 2014; Wang, Paisley, & Blei, 2011).

Model choice is typically based at least partially on subjective considerations similar to

those in the qualitative research (Grimmer & Stewart, 2013). These six topics represented

more nuanced and conceptually meaningful topics that belonged to the two cross-cutting

‘meta-topics’: ‘ECE finance’ and ‘ECE services.’

Having determined a final set of six policy topics, we further elaborate on the

process of assigning the meaning of the topics and assessing their content validity. Topic

validation becomes essential for automated text analysis since the LDA algorithm’s

underlying frequency analysis may identify ‘word groupings’ as ‘topics’, however, these

‘word groupings’ may not strike a human observer as having a semantic structure. To

ensure computer-generated topics captured topics that were theoretically and practically

meaningful, two expert coders evaluated the content validity and semantic coherence of

the six topics. Content validity denotes the extent to which the topics identify coherent

and distinct sets of ECE policy priorities and measure conceptually sound constructs.

Having expert coders who have relevant theoretical and practical knowledge was critical

for assessing topics’ content validity (Sun et al, 2019). Figure 5 displays a unique set of

75 words with highest probabilities of appearing in each topic and that best distinguish

the topics from one another. Using these high probability terms, we independently coded

and labeled each of the six topics and assessed its coherence (how these words hold

together as a construct) and the extent to which a given topic is meaningful and consistent

with the literature. To further aid this process, we carefully read high probability terms in

original legislative text, inferring its definition, meaning, and usage and their relation to

the topic. Upon repeating this process for each model specification, we compared and

discussed our qualitative coding decisions. We reached agreement on classifying most of

the topics, and refined the label for just one topic, ‘Fiscal Governance,’ that integrated

both the administration and financing of ECE services.

For ECE finance, we identified three specific topics -- ‘revenues’, ‘expenditures’,

and ‘fiscal governance’. Unique keywords with highest probabilities of appearing under

‘revenues’ were adjustment (of schedule to facilitate department accounting or unit

allocation), agreement (i.e., property tax or tobacco tax agreement), a llocation, allowable

tax credit, allowance (for instructional classroom support) business, calculation,

corporation, credit, distribute, dollar, income, fiscal intermediary, properties, residence,

sale, tax, taxable, taxpayers . These words together suggested budgetary sources needed

to finance early childhood care and educational services, particularly in schools and

districts as seen in Figure 2. High frequency words for ‘expenditures’ included benefit,

care, children, clients, disabled children, families, hospital, maltreatment, medical,

medicaid, mental health, nurse and treatment , suggesting what it would cost for various

providers in delivering early childhood services including mental health and social

welfare services for direct beneficiaries. Furthermore, several keywords, for example,

indicated expenditures in terms of financing the early childhood workforce (i.e.,

structuring salaries and benefits). This process involves setting standards for teacher

qualifications and other structural characteristics (e.g., class size, ratio, hours per week)

of programs that affect cost. Since public expenditures on state preschool come from

federal, state as well as local sources, states vary in their program schedule (half-day or a

full-day) and leave the schedule up to local discretion (Barnett et al., 2009). Local

authorities determine how thinly state funding is stretched regarding length of day and

frequently, data are unavailable about how many children are served with each type of

schedule. In managing state budget--both revenues and expenditures--the third

finance-related topic emerged as ‘fiscal governance’ dealing with account,

appropriation, balance, capital (capital gain, capital expenditure, capital loss) ,

commission, committee, contract, finance, item, local, money, officials, reimbursement,

salaries, schedule, fiscal transfer (by administrative bodies to accounts) , and trust

(governing board of trustees as well as trust management). These words seemed to

suggest the logistics of locating the money to pay for the programs at the legislative level

and facilitating efficient distribution and use of resources. A commonly shared set of

words that appeared across the three topics further confirmed ECE finance as the

underlying theme-- amount, authorize, communities, cost, counties, department, fund,

payment, provide, and public.

For ECE services, three specific topics included governance and provision of

early childhood services: ‘prekindergarten (PreK),’ ‘child care,’ and ‘health and human

services (HHS).’ We found that three key topics--‘prekindergarten’ (preK), ‘child care’,

and ‘health and human services’ (HHS)--addressed policy questions of what (types and

quality of provision, program components, and dosage), for whom (targeted population,

access and participation trends particularly among historically marginalized children and

their families), where (delivery settings), as well as by whom (administration leaders and

providers). Classification of these three topics closely reflect the complexity of early childhood governance as well as the fragmentation and duplication of federal funding

streams and state administration across early childhood programs and services. More

specifically, unique words with highest probabilities of appearing for ‘preK’ included

academic, achieve(ment), charter, college-ready, commission(er), district, enroll, grade,

instruct, kindergarten, prekindergarten, preschool, pupil, school, student, and teacher . In

comparison, high frequency words under ‘child care’ included assist(ing), children

(rather than pupils or students under ‘preK’ topic), counties, daycare, income, infant,

subsidi(-es, -ze), and home visit . These two topics, ‘preK’ and ‘child care’, mutually

focused on promoting the development and early learning opportunities for eligible

children through educational programming and services. Unlike the topics ‘preK’ and

‘child care’, ‘HHS’ embodied a broader system of early childhood and social welfare

services as suggested by words such as a buse, adopt(ion), agency, child care, convict(-ed,

-ion), court, criminal, daycare, foster care, health, home visit, neglect, personnel, for

example. The theory of change undergirding ‘HHS’ services posits that preparing

children for success in school and life involve not only their cognitive development and

academic enrichment but also their physical and mental health as well as social-emotional

well-being and family support services. Types of services may include home visits,

parent support or training, referral to social services, and health services. Additionally,

examining a commonly shared set of words underlying these three topics further

supported our classification of these topics having to do with ECE programs and services:

Words such as authorize, children, parents, program, provide, public, purpose, receive,

require and services further supported our conceptualization of these three topics as

complementary yet distinct dimensions of providing public early childhood programs and

services that are designed to promote children’s learning and healthy development.

Reviewer #1c: Authors mentioned the multilevel structure of the data and used HGLM. I am just

wondering whether LDA is OK for the multilevel data structure. Further, in general, is it ok to

apply machine learning approaches such as LDA to multilevel data structure? Why?

RESPONSE : Thank you for raising this. We have addressed your question in the analytic

strategy section for RQ2 as seen below.

“ If OLS was used for this type of data, the statistical significance of the regression results

might be overstated, an issue which is not relevant in the LDA itself since statistical

hypothesis testing is not being done for the topics themselves.”

We have tried to avoid a lengthy treatment as this is an applied paper. Hence, we have opted to

respond in detail below:

To preface the actual answer, let us review some fundamentals. Statistics is principally

concerned with two overarching pursuits-- statistical inference and prediction . Multilevel structure

has been identified as a potential issue in statistical inference as it is the part of statistics that is

concerned with hypothesis testing, whereas prediction chiefly uses cross-validation (in effect, an

out-of-sample prediction performance) to ascertain its suitability for the goal at hand. In

statistical inference, we care about the ‘beta’ (quantifiers of the association between the

outcome and independent variables) coefficients, about them being unbiased estimates of the

population quantities in question, and about the standard error associated with the estimator

being also unbiased. In particular, the issue with hierarchical modeling is that two observations

from the same state (in this case) would not bring as much information as two observations from

different states and that treating them as such might inflate the null hypothesis rejection rate in

our analysis (find an effect too often). This is why in the inference section of our paper (RQ2 &

RQ3), we have opted for HLM as we wanted to be conservative with our hypothesis testing and

statistical inference. Another way of looking at this is that being conservative in inference means

using a better model (with more realistic assumptions), while being conservative in prediction

means using a ‘worse’ (a more simple) model (to avoid overfitting the training data and being

less generalizable as a result).

The LDA model (RQ1) in our analysis serves as a prediction model, in effect synthesizing a

variable later to be used as an independent variable in an inference model (RQ2 & RQ3). Let us

play out a hypothetical scenario where we use an LDA in RQ1 and some other model B (could

be a better version of LDA with incorporation of states or another model altogether) to identify

topics. Let us assume that model B performs ‘better’ in a sense that the topics it discovers are

clear-cut and contain fewer words that should really belong to another topic -- that it introduces

a clean (as opposed to ‘blurry’) classification of topics in the legislation. Then for the inference

purposes (RQ2 & RQ3) we would say that model B gives predictors that are less noisy. This will

affect the inference in RQ2 & RQ3 via lower standard errors on the ‘beta’ coefficients associated

with those topics. This, in turn, will make it more likely that those topics are statistically

significant predictors of the legislation’s success. To restate what we mentioned above, in

inference, better modeling such HLM can sometimes nullify results identified with simpler

models such as OLS. In prediction, better modeling does not nullify results.

A noisy predictor would make it harder to reject the null of no effect of topics on legislative

success. In this sense, the reason why we are mindful of hierarchical structure in the data with

RQ2 and RQ3 is because we are conservative, while the reason why this is not as much of an

issue with RQ1 is that we only need a model ‘good enough’ to discover topics that in our

disciplinary knowledge and understanding of the early childhood field appear a reasonable

reflection of U.S. early childhood policies. Had we incorporated states in some sense as

information into the LDA model, we could have increased the topic clarity and rejected the null

of no association with even greater confidence in RQ2 & RQ3, but the conclusion would remain

unaltered. With a better natural text processing model, the association between legislation’s

success and topics would be even stronger. Finally, by demonstrating the relevance of topics

with even a bare-bones NLP model, we are offering the research community a lower bar on

what can be achieved in a continued pursuit of NLP algorithms in application to early childhood

policy analysis. Given the NLP algorithms are still an active area of research, our paper gives

the ECE community a preview and a lower bar on the benefits that come with it.

A more detailed treatment of the issues discussed above can be found in:

● Hastie, T., Tibshirani, R., & Friedman, J. (2009). The elements of statistical learning: data mining,

inference, and prediction . Springer Science & Business Media.

● Gelman, A., & Hill, J. (2006). Data analysis using regression and multilevel/hierarchical models .

Cambridge university press.

Reviewer #2 : This paper presents an original contribution to the field through an analytic

approach implementing machine-learning to unlock the potential of legislative data to inform

future policymaking in early childhood-care and education.

The rationale for the research question is clearly stated. The general approach and statistical

analyses were described and carried out rigorously. The discussion and conclusion

appropriately derive from the data presented.

Although "the use of big data in research on public policies has lagged behind other fields", it is

important to highlight some recent studies which applied machine-learning to predict

educational outcomes at large scale and/or educational data mining field (for example: Musso

MF, Cascallar EC, Bostani N and Crawford M (2020) Identifying Reliable Predictors of

Educational Outcomes Through Machine-Learning Predictive Modeling. Frontiers in Education.

5:104. doi: 10.3389/feduc.2020.00104, and Romero, C & Ventura, S. (2017) Educational Data

Science in Massive Open Online Courses. Wiley Interdisciplinary Reviews: Data Mining and

Knowledge Discovery · January 2017).

RESPONSE : We appreciate Reviewer #2 for sharing these relevant references and specifying

where we might cite them. We agree that highlighting such emerging studies fairly

acknowledges emerging efforts in the educational data mining field and further enriches our

discussion of findings and future directions for research. We have incorporated your suggested

sources along with others to highlight emerging applications of topic modeling in educational

research both in the literature review and the discussion section of our revised manuscript (see

revised excerpts below):

“Topic Modeling (Blei et al., 2003) is a rapidly emerging area of research in

educational data mining. Text analysis methods in education research have been used to

1) measure latent dispositions such as attitudes and beliefs of learners and instructors

(U.S. Department of Education, 2012); 2) explore the underlying topics and topic

evolution spanning the 50-year history of educational leadership research literature

(Wang et al., 2017); 3) microclassroom processes such as MOOC interaction data

(Romero & Ventura, 2017); 4) policy implementation and reform strategies (Sun et al.,

2019); or 5) constructing artificial neural network models of multiple setting-level

predictors of student’s language and math achievement outcomes (Musso et al., 2020).

Despite the promising potential of applying topic modeling in a variety of fields in the

social sciences, its scalable, algorithmic approach to large-scale text data has received

little attention among early childhood and education policy scholars (Wang et al., 2017).”

“While the use of big data broadly in research on public policies and

publicly-funded programs has lagged behind other fields such as computer science and

medicine (Coulton et al., 2015), LDA in particular has been utilized in understanding

legislative policymaking (Blei et al., 2003; Nay 2017). Additionally, the application of

computer-assisted analyses of large-scale text such as text reuse methods (Wilkerson et

al., 2015), topic modeling (Blei et al., 2003) and classification methods are far sparse in

education policy research (Sun et al., 2019). However, a burgeoning body of work has

applied machine learning algorithms in educational research to better understand how

teaching and learning can be enhanced for whom under what conditions (Elatia et al.,

2016; U.S. Department of Education, 2012). Recent studies, for example, have built

artificial neural network models that predict students’ language and math performance at

large scale (Musso et al., 2020), introduced various methods in educational data science

(EDS) for examining students’ massive open online courses (MOOCs) interactions

(Romero & Ventura, 2017), or using LDA to analyze text data from thousands of school

improvement reports to identify reform mechanisms that reduced student chronic

absenteeism and improved achievement (Sun et al., 2019).”

---

## [Decision Letter · Decision Letter 1]

26 Jan 2021

What predicts legislative success of Early Care and Education (ECE) policies?: Applications of machine learning and natural language processing in a cross-state early childhood policy analysis

PONE-D-20-13754R1

Dear Dr. Park,

We’re pleased to inform you that your manuscript has been judged scientifically suitable for publication and will be formally accepted for publication once it meets all outstanding technical requirements.

Kind regards,

Mingming Zhou, Ph.D.

Academic Editor

PLOS ONE

Additional Editor Comments (optional):

Reviewers' comments:

Reviewer's Responses to Questions

**Comments to the Author**

1. If the authors have adequately addressed your comments raised in a previous round of review and you feel that this manuscript is now acceptable for publication, you may indicate that here to bypass the “Comments to the Author” section, enter your conflict of interest statement in the “Confidential to Editor” section, and submit your "Accept" recommendation.

Reviewer #2: All comments have been addressed

2. Is the manuscript technically sound, and do the data support the conclusions?

Reviewer #2: Yes

3. Has the statistical analysis been performed appropriately and rigorously? 

Reviewer #2: Yes

4. Have the authors made all data underlying the findings in their manuscript fully available?

Reviewer #2: Yes

5. Is the manuscript presented in an intelligible fashion and written in standard English?

Reviewer #2: Yes

6. Review Comments to the Author

Reviewer #2: The authors have answered exhaustively all the reviewers' comments and they have included the suggestions.

The manuscript provides an original contribution to the field and the study was carried out rigorously.

7. PLOS authors have the option to publish the peer review history of their article (what does this mean?). If published, this will include your full peer review and any attached files.

Reviewer #2: No

---

## [Editor Report · Acceptance letter]

2 Feb 2021

PONE-D-20-13754R1 

What Predicts Legislative Success of Early Care and Education Policies?: Applications of Machine Learning and Natural Language Processing in A Cross-State Early Childhood Policy Analysis  

Dear Dr. Park:

I'm pleased to inform you that your manuscript has been deemed suitable for publication in PLOS ONE. Congratulations! Your manuscript is now with our production department. 

Kind regards, 

on behalf of

Dr. Mingming Zhou 

Academic Editor

PLOS ONE